# Dynamic Tensor Product Regression

**Aravind Reddy**[*]    **Zhao Song**[†]    **Lichen Zhang**[‡]

## Abstract

In this work, we initiate the study of *Dynamic Tensor Product Regression*. One has matrices $A_1 \in \mathbb{R}^{n_1 \times d_1}, \ldots, A_q \in \mathbb{R}^{n_q \times d_q}$ and a label vector $b \in \mathbb{R}^{n_1 \cdots n_q}$, and the goal is to solve the regression problem with the design matrix $A$ being the tensor product of the matrices $A_1, A_2, \ldots, A_q$ i.e. $\min_{x \in \mathbb{R}^{d_1 \cdots d_q}} \|(A_1 \otimes \ldots \otimes A_q)x - b\|_2$. At each time step, one matrix $A_i$ receives a sparse change, and the goal is to maintain a sketch of the tensor product $A_1 \otimes \ldots \otimes A_q$ so that the regression solution can be updated quickly. Recomputing the solution from scratch for each round is very slow and so it is important to develop algorithms which can quickly update the solution with the new design matrix. Our main result is a dynamic tree data structure where any update to a single matrix can be propagated quickly throughout the tree. We show that our data structure can be used to solve dynamic versions of not only Tensor Product Regression, but also Tensor Product Spline regression (which is a generalization of ridge regression) and for maintaining Low Rank Approximations for the tensor product.

## 1 Introduction

The task of fitting data points to a line is well-known as the least-squares regression problem [Sti81], which has a wide range of applications in signal processing [RGlY78], convex optimization [Bub15], network routing [Mad13, Mad16], and training neural networks [BPSW21, SZZ21]. In this work, we study a generalized version of least-squares regression where the design matrix $A$ is a tensor product of $q$ smaller matrices $A_1 \otimes A_2 \otimes \ldots \otimes A_q$.

Tensor products have been extensively studied in mathematics and the physical sciences since their introduction more than a century ago by Whitehead and Russell in their Principia Mathematica [WR12]. They have been shown to have a humongous number of applications in several areas of mathematics like applied linear algebra and statistics [VL00]. In particular, machine learning applications include image processing [NK06], multivariate data fitting [GVL13], and natural language processing [PSA21]. Furthermore, regression problems involving tensor products arise in surface fitting and multidimensional density smoothing [EM06], and structured blind deconvolution problems [OY05], among several other applications, as discussed in [DSSW18, DJS$^+$19].

Solving tensor product regression in $\ell_2$ norm [DJS$^+$19, FFG22] to a $(1 \pm \varepsilon)$ precision takes time $\widetilde{O}(\sum_{i=1}^{q} \mathrm{nnz}(A_i) + \mathrm{poly}(dq/(\delta\varepsilon)))$, where each matrix $A_i \in \mathbb{R}^{n_i \times d_i}$ and $d = d_1 d_2 \ldots d_q$, and $\mathrm{nnz}(A)$ denotes the number of nonzero entries of $A$. However, these algorithms are inherently *static*, meaning that even if there is a sparse (or low-rank) update to a *single* design matrix $A_i$, it needs to completely recompute the new solution from scratch. In modern machine learning applications, such static algorithms are not practical due to the fact that data points are always evolving and recomputing the solution from scratch every time is computationally too expensive. Hence, it is important to develop efficient *dynamic algorithms* that can adaptively update the solution. For example, graphs for real-world network data are modeled as the tensor product of a large number of

---

[*]`aravind.reddy@cs.northwestern.edu`. Northwestern University.

[†]`zsong@adobe.com`. Adobe Research.

[‡]`lichenz@mit.edu`. MIT. (Author names in alphabetical order, equal contribution)

36th Conference on Neural Information Processing Systems (NeurIPS 2022).

smaller graphs [LCK$^+$10]. Many important problems on these graphs can be solved by regression of the adjacency and Laplacian matrices of these graphs [ST04]. Real-world network data is always time-evolving and so it is crucial to develop dynamic algorithms for solving regression problems where the design matrix is a tensor product of a large number of smaller matrices. Hence, we ask the following question:

*Can we design a dynamic data structure for tensor product regression that can handle updates to the design matrix and quickly updates the solution?*

We provide a positive answer to the above question.

At the heart of our data structure is a binary tree that can maintain a succinct sketch of $A_1 \otimes \ldots \otimes A_q$ and supports an update to one of the matrices quickly. Such tree structures have been useful in designing efficient sketching algorithms for tensor products of vectors [AKK$^+$20, SWYZ21]. However, the goal of these works has been in reducing the sketching dimension from an exponential dependence on $q$ to a polynomial dependence on $q$. Their results can be directly generalized to solving tensor product regression *statically* but not dynamically. In this work, we build upon the tree structure to solve the Dynamic Tensor Product Regression problem and other related dynamic problems like Dynamic Tensor Spline Regression and Dynamic Tensor Low Rank Approximation. Our key observation is that updating the entire tree is efficient when a single leaf of the tree gets an update: specifically, we only need to update the nodes which fall on the path from the leaf to the root.

**Technical Contributions.**

- We design a dynamic tree data structure DYNAMICTENSORTREE that maintains a succinct representation of the tensor product $A_1 \otimes \ldots \otimes A_q$ and supports efficient updates to any of the $A_i$'s.

- Consequently, we develop a dynamic algorithm for solving Tensor Product Regression, when one of the matrices $A_i$'s is updated and we need to output an estimate of the new solution to the regression problem quickly.

- We also show that we can use our DYNAMICTENSORTREE data-structure in a fast dynamic algorithm to solve the Tensor Product Spline Regression problem, which is a generalization of the classic Ridge Regression problem.

- We also initiate the study of *Dynamic Tensor Low Rank Approximation*, where the goal is to maintain a low rank approximation (LRA) of the tensor product with dynamic updates, and show how we can use our DYNAMICTENSORTREE data structure to solve this problem.

**Roadmap.** In section 2, we first discuss some related work. Then, we provide notation, some background for our work, and the main problem formulation in section 3. In section 4, we provide a technical overview of our paper. Following that, we provide our dynamic tree data structure in section 5. We then show how it can be used for Dynamic Tensor Product Regression, Dynamic Tensor Spline Regression, and Dynamic Tensor Low Rank Approximation in section 6. We end with our conclusion and discuss some future directions in section 7.

## 2 Related Work

**Sketching.** Sketching techniques have many applications in numerical linear algebra, such as linear regression, low-rank approximation [CW13, NN13, MM13, BW14, RSW16, SWZ17, HLW17, ALS$^+$18, BBB$^+$19, IVWW19, SWZ19a, SWZ19b, MW21, WY22, CSTZ22], distributed problems [WZ16, BWZ16], reinforcement learning [WZD$^+$20, SSX21], projected gradient descent [XSS21], training over-parameterized neural networks [SYZ21, SZZ21], tensor decomposition [SWZ19c], clustering [EMZ21], cutting plane method [JLSW20], generative adversarial networks [XZZ18], recommender systems [RRS$^+$22], and linear programming [LSZ19, JSWZ21, SY21].

**Dynamic Least-squares Regression.** A dynamic version of the ordinary least-squares regression problem (without a tensor product design matrix) was recently studied in [JPW22]. In their model, at each iteration, a new row is prepended to the design matrix $A$ and a new coordinate is prepended

to the vector $b$. To efficiently maintain the necessary parts of the design matrix and the solution, they make use of a fast leverage score maintenance data structure by [CMP20] and achieve a running time of $\widetilde{O}(\mathrm{nnz}(A^{(T)}) + \mathrm{poly}(d, \varepsilon^{-1}))$ where $T$ is the number of time steps.

**Tensor/Kronecker Product Problems.** Many machine learning problems involve tensor/Kronecker product computations such as, regression [HLW17, DSSW18, DJS$^+$19], low-rank approximation [SWZ19c], fast kernel computation [SWYZ21], semi-definite programming [JKL$^+$20, HJS$^+$21], and training over-parameterized neural networks [BPSW21, SZZ21]. Apart from solving problems which directly involve tensor/Kronecker products, another line of research focuses on constructing polynomial kernels efficiently so that the computation of various kernel problems can be improved [AKK$^+$20, SWYZ21].

# 3 Preliminaries

## 3.1 Notation

For any natural number $n$, we use $[n]$ to denote the set $\{1, 2, \ldots, n\}$. We use $\mathbb{E}[\cdot]$ to denote expectation of a random variable if it exists. We use $\mathbf{1}[\cdot]$ and $\Pr[\cdot]$ to denote the indicator function and probability of an event respectively. For any natural numbers $a, b, c$, we use $\mathcal{T}_{\mathrm{mat}}(a, b, c)$ to denote the time it takes to multiply two matrices of sizes $a \times b$ and $b \times c$. For a vector $x$, we use $\|x\|_2$ to denote its $\ell_2$ norm. For a matrix $A$, we use $\|A\|_F$ to denote its Frobenius norm. For a matrix $A$, we use $A^\top$ to denote the transpose of $A$. Given two matrices $A \in \mathbb{R}^{n_1 \times d_1}$ and $B \in \mathbb{R}^{n_2 \times d_2}$, we use $A \otimes B$ to denote their tensor product (which is also referred to as Kronecker product), i.e., $(A \otimes B)_{i_1 + (i_2-1) \cdot n_1, j_1 + (j_2-1) \cdot d_1} = A_{i_1, j_1} \cdot B_{i_2, j_2}$. For example, for $2 \times 2$ matrices $A$ and $B$,

$$\begin{bmatrix} a_{11} & a_{12} \\ a_{21} & a_{22} \end{bmatrix} \otimes B = \begin{bmatrix} a_{11}B & a_{12}B \\ a_{21}B & a_{22}B \end{bmatrix} = \begin{bmatrix} a_{11}b_{11} & a_{11}b_{12} & a_{12}b_{11} & a_{12}b_{12} \\ a_{11}b_{21} & a_{11}b_{22} & a_{12}b_{21} & a_{12}b_{22} \\ a_{21}b_{11} & a_{21}b_{12} & a_{22}b_{11} & a_{22}b_{12} \\ a_{21}b_{21} & a_{21}b_{22} & a_{22}b_{21} & a_{22}b_{22} \end{bmatrix}$$

We use $\bigotimes$ to denote tensor product of more than two matrices, i.e., $\bigotimes_{i=1}^{q} A_i = A_1 \otimes A_2 \otimes \cdots \otimes A_q$. For non-negative real numbers $a$ and $b$, we use $a = (1 \pm \varepsilon)b$ to denote $(1 - \varepsilon) \cdot b \leq a \leq (1 + \varepsilon) \cdot b$. For a real matrix $A$, we use $\sigma_i(A)$ to denote its $i$-th largest singular value. For any function $f$, we use $\widetilde{O}(f)$ to denote $O(f \mathrm{poly}(\log f))$.

## 3.2 Problem Formulation

In this section, we define our task. We start with the *static version* of tensor product regression.

**Definition 3.1** (Static version). In the $\ell_2$ Tensor Product Regression problem, we are given matrices $A_1, A_2, \ldots, A_q$ where $A_i \in \mathbb{R}^{n_i \times d_i}$ and $b \in \mathbb{R}^{n_1 n_2 \ldots n_q}$. Let us define $n := n_1 n_2 \ldots n_q$ and $d := d_1 d_2 \ldots d_q$. The goal is to output $x \in \mathbb{R}^d$ such that we minimize the objective:

$$\|(A_1 \otimes A_2 \otimes \cdots \otimes A_q)x - b\|_2$$

Our dynamic version of $\ell_2$ Tensor Product Regression involves updating one of the matrices $A_i$ with an update matrix $B$. We formally define the problem as follows:

**Definition 3.2** (Dynamic version). In the Dynamic $\ell_2$ Tensor Product Regression problem, we are given matrices $A_1, A_2, \ldots, A_q$ where $A_i \in \mathbb{R}^{n_i \times d_i}$, a sequence $\{(i_1, B_1), (i_2, B_2), \ldots, (i_T, B_T)\}$ with $i_t \in [q], B_t \in \mathbb{R}^{n_{i_t} \times d_{i_t}}$ for all $t \in [T]$, and the goal is to design a data structure with the following procedures:

- INITIALIZE: The data structure initializes $A_1, \ldots, A_q$ and maintains an approximation to $\bigotimes_{i=1}^{q} A_i$.

- UPDATE: At time step $t$, given an index $i_t$ and an update matrix $B_t$, the data-structure needs to update $A_{i_t} \leftarrow A_{i_t} + B_t$, and maintain an approximation to $A_1 \otimes \ldots \otimes (A_{i_t} + B_t) \otimes \ldots \otimes A_q$.

- QUERY: The data structure outputs a vector $\widehat{x} \in \mathbb{R}^d$ such that

$$\|(\bigotimes_{i=1}^{q} A_i)\widehat{x} - b\|_2 = (1 \pm \varepsilon) \min_{x \in \mathbb{R}^d} \|(\bigotimes_{i=1}^{q} A_i)x - b\|_2.$$

Note that although we don't discuss updates to the vector $b$ in our model, they are easy to implement in our data-structure. In addition to the standard regression problem, we also consider the spline regression (which is a generalization of ridge regression) and low rank approximation problems. We will provide the definitions for these problems in their respective sections.

## 4 Technical Overview

The two main design considerations for our data structure are as follows: 1). We want a data structure that stores a *sketch* of the tensor product, this has the advantage of having a smaller storage space and a faster time to form the tensor product. 2). The update time to one of the matrices $A_i$ should be fast, ideally we want to remove the linear dependence on $q$. With these two goals in mind, we introduce a dynamic tree data structure that satisfies both of them simultaneously, building on the sketch of [AKK+20]. Specifically, we use each leaf of the tree to represent $S_i A_i$, a sketch of the base matrix that only has $m$ rows instead of $n$ rows, where $m$ is proportional to the small dimension $d$. Each internal node represents a sketch of the Tensor product of its two children. To speed up computation, we adapt tensor-typed sketching matrices to avoid the need to explicitly form the Tensor product of the two children. The root of the tree will store a sketch for the tensor product $\bigotimes_{i=1}^{q} A_i$.

We point out that in order for this tree to work, we need to apply an *independent* sketching matrix to each tree node. Since there are $2q - 1$ nodes in total, we will still have a dependence on $q$ in the initialization phase (which will be dominated by the $d$ term). However, during the update phase, our tree has a clear advantage: if one of the leaves has been updated, we only need to propagate the change along the path to root. Since the height of the tree is at most $O(\log q)$, we remove the linear dependence on $q$ in the update phase. This is the key insight for obtaining a faster update time using our dynamic data structure. Moreover, the correctness follows naturally from the guarantee of the tree: as shown in [AKK+20], the tree itself is an Oblivious Subspace Embedding (OSE) for matrices of proper size, hence, it also preserves the subspace after updating one of the leaves. Figure 1 is an example tree for tensor product of 4 matrices.

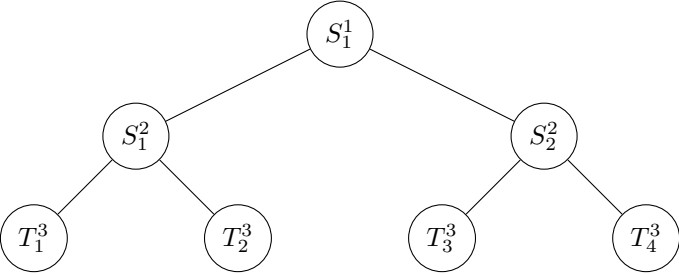

Figure 1: A tree for sketching the tensor product of 4 matrices $A_1, A_2, A_3$ and $A_4$. The leaves use sketches of type $T_{\mathrm{base}}$ and internal nodes use sketches of type $S_{\mathrm{base}}$. The algorithm starts to apply $T_i$ to each matrix $A_i$, and then propagates them up the tree.

With this data structure, we provide efficient algorithms for dynamic Tensor product regression, dynamic Tensor Spline regression, and dynamic Tensor low rank approximation. For all these problems, we do not explicitly maintain the solutions, rather, we use our tree to quickly update a sketch of the design matrix. In the query phase (when we need to actually solve the regression problems), we use this sketch of updated design matrix to quickly compute the solution.

## 5 Dynamic Tree Data Structure

In this section, we introduce our data structure, which contains INITIALIZE and UPDATE procedures. We'll define task-specific QUERY procedures later. We defer proofs in this section to Appendix B. We start with an outline of Algorithm 1.

---

**Algorithm 1** Our dynamic tree data structure

---

1: **data structure** DYNAMICTENSORTREE                                    ▷ Theorem 5.1
2:
3: **members**
4:     $J_{k,\ell} \in \mathbb{R}^{m \times d^{2^\ell}}$ for $0 \le \ell \le \log q$ and $0 \le k \le q/2^\ell - 1$            ▷ $\ell$ is the level of the tree.
5: **end members**
6:
7: **procedure** INITIALIZE($A_1 \in \mathbb{R}^{n_1 \times d_1}, \ldots, A_q \in \mathbb{R}^{n_q \times d_q}, m \in \mathbb{N}_+, C_{\text{base}}, T_{\text{base}}$)
8:     **for** $\ell = 0 \to \log q$ **do**
9:         **for** $k = 0 \to q/2^\ell - 1$ **do**
10:            **if** $\ell = 0$ **then**
11:                Choose a $C_k \in \mathbb{R}^{m \times n_k}$ from $C_{\text{base}}$. ▷ $C_{\text{base}}$ can be COUNTSKETCH, SRHT or OSNAP.
12:                $J_{k,\ell} \leftarrow C_k A_k$
13:            **else**
14:                Choose a $T_{k,\ell} \in \mathbb{R}^{m \times m^2}$ from $T_{\text{base}}$.            ▷ $T_{\text{base}}$ can be TENSORSKETCH or TENSORSRHT.
15:                $J_{k,\ell} \leftarrow T_{k,\ell}(J_{2k,\ell-1} \otimes J_{2k+1,\ell-1})$      ▷ Sketch of tensor product of its children
16:            **end if**
17:        **end for**
18:    **end for**
19: **end procedure**
20:
21: **procedure** UPDATE($i \in [q], B \in \mathbb{R}^{n_i \times d_i}$)
22:     **for** $\ell = 0 \to \log(q)$ **do**
23:        **if** $\ell = 0$ **then**
24:            $J_{i,0}^{\text{new}} \leftarrow C_i B$
25:            $J_{i,0} \leftarrow J_{i,0} + J_{i,0}^{\text{new}}$
26:        **else**
27:            $J_{\lceil i/2^\ell \rceil, \ell}^{\text{new}} \leftarrow T_{k,\ell}(J_{\lceil i/2^{\ell-1} \rceil, \ell-1}^{\text{new}} \otimes J_{\lceil i/2^{\ell-1} \rceil + 1, \ell-1})$            ▷ Sibling is to the left also sometimes
28:            $J_{\lceil i/2^\ell \rceil, \ell} \leftarrow J_{\lceil i/2^\ell \rceil, \ell} + J_{\lceil i/2^\ell \rceil, \ell}^{\text{new}}$            ▷ Sketch of tensor product of its children
29:        **end if**
30:    **end for**
31: **end procedure**

---

**Outline of Tree:** For simplicity, let us assume that the number of matrices $q$ is a power of 2. Also, let us assume that all the matrices are of the same dimension $n_1 \times d_1$. The output is a matrix of dimension $m \times d$ where $m = \text{poly}(q, d, \varepsilon^{-2})$. First, we apply $q$ independent base sketch matrices (such as COUNTSKETCH (Def. A.6), SRHT (Def. A.8)) $\{T_i \in \mathbb{R}^{m \times n_1}, i \in [q]\}$ to each of the matrices $A_1, A_2, \ldots, A_q$. After this step, all the leaf nodes are of size $m \times d_1$. Then, we have a tree of height $\log q$. Let us use $\ell = 0 \to \log q$ to represent the level of the tree nodes where $\ell = 0$ represents the leaves and $\ell = \log q$ represents the root node. Let us use $J_{k,\ell} \in \mathbb{R}^{n \times d^{2^\ell}}$ to denote the matrix stored at the node $k, \ell$. For leaf nodes, we choose base sketches $T_k \in \mathbb{R}^{m \times n_1}$ and compute $J_{k,0} = T_k A_k$. At each internal node, we use a sketch for tensor-typed inputs (such as TENSORSKETCH (Def. A.7) or TENSORSRHT (Def. A.5)), apply it to its two children. The sketching matrix $S_i \in \mathbb{R}^{m \times m^2}$, and it can be applied to its two inputs fast, obtaining a running time of $\widetilde{O}(m)$ instead of $O(m^2)$. We inductively compute the tree all the way up to the root for initialization. Note that at the root, we end up with a matrix of size $m \times d$, which is significantly smaller than the tensor product $n \times d$.

The running time guarantees of Algorithm 1, which we will prove in Appendix B, are as follows:

**Theorem 5.1** (Running time of our main result. Informal version of Theorem B.1). *There is an algorithm (Algorithm 1) that has the following procedures:*

- INITIALIZE$(A_1 \in \mathbb{R}^{n_1 \times d_1}, A_2 \in \mathbb{R}^{n_2 \times d_2}, \ldots, A_q \in \mathbb{R}^{n_q \times d_q}, m \in \mathbb{N}_+, C_{\text{base}}, T_{\text{base}})$: *Given matrices $A_1 \in \mathbb{R}^{n_1 \times d_1}, A_2 \in \mathbb{R}^{n_2 \times d_2}, \ldots, A_q \in \mathbb{R}^{n_q \times d_q}$,, a sketching dimension $m$, families of base sketches $C_{\text{base}}$ and $T_{\text{base}}$, the data-structure pre-processes in time $\widetilde{O}(\sum_{i=1}^{q} \text{nnz}(A_i) + qmd)$.*

- UPDATE$(i \in [q], B \in \mathbb{R}^{n_i \times d_i})$: *Given a matrix $B$ and an index $i \in [q]$, the data structure updates the approximation to $A_1 \otimes \ldots \otimes (A_i + B) \otimes \ldots \otimes A_q$ in time $\widetilde{O}(\text{nnz}(B) + md)$.*

**Remark 5.2.** In our data structure, we only pay the linear $q$ term in INITIALIZATION (which will be dominated by the term $d$), since we are maintaining a tree with $2q - 1$ nodes, each node corresponds to a matrix of $m$ rows. In the UPDATE phase, we shave off the linear dependence on $q$. We also want to point out that the sketching dimension $m$ is typically at least linear on $q$, however, it only depends on $q$, the small dimension $d$ and the precision parameter $\varepsilon$, hence it is far more efficient to use this sketch representation compared to the original tensor product. Also, note that though we only design a data structure for $q$ being a power of 2, it is not hard to generalize to $q$ being an arbitrary positive integer via the technique introduced in [SWYZ21]. We only incur a $\log q$ factor by doing so.

The approximation guarantee for our dynamic tree, which follows from [AKK$^+$20] and for which we provide a brief sketch in Appendix B are as follows:

**Theorem 5.3** (Approximation guarantee of our dynamic tree. Informal version of Theorem B.1)**.** *Let $\Pi^q$ denote the sketching matrix generated by the dynamic tree, we then have that:*

$$\|\Pi^q(\bigotimes_{i=1}^{q} A_i)x\|_2 = (1 \pm \varepsilon)\|(\bigotimes_{i=1}^{q} A_i)x\|_2.$$

**Choices of parameter $m$:** An important parameter to be chosen is the sketching dimension $m$. Intuitively, it should depend polynomially on $\varepsilon^{-1}, d, q$. We list them in the following table.

| $C_{\text{base}}$ | $T_{\text{base}}$ | $m(\dim)$ | **Init time** |
|---|---|---|---|
| COUNTSKETCH | TENSORSKETCH | $\varepsilon^{-2}q \cdot \dim^2 \cdot (1/\delta)$ | $\sum_{i=1}^{q} \text{nnz}(A_i)$ |
| OSNAP | TENSORSRHT | $\varepsilon^{-2}q^4 \cdot \dim \cdot \log(1/\delta)$ | $\sum_{i=1}^{q} \text{nnz}(A_i)$ |
| OSNAP | TENSORSRHT | $\varepsilon^{-2}q \cdot \dim^2 \cdot \log(1/\delta)$ | $\sum_{i=1}^{q} \text{nnz}(A_i)$ |

Table 1: How different choices of $C_{\text{base}}$ and $T_{\text{base}}$ affect the sketching dimension $m(\dim)$ and initialization time that depends on the choice of sketch. We note that the sketching dimension $m$ is a function of $\dim$, the fundamental dimension of the problem. For example, $\dim = d$ for tensor product regression and $\dim = k$ for $k$-low rank approximation. The sketching dimension corresponds to the problem of computing an $(\varepsilon, \delta, \dim, d, n)$-OSE.

**Robustness against an Adaptive Adversary:** An often encountered issue in dynamic algorithms is the presence of an *adaptive adversary* [BKM$^+$22]. Consider a sequence of update pairs $\{(i, B_i)\}_{i=1}^{T}$ in which the adversary can choose the pair $(i, B_i)$ based on the output of our data structure on $(i - 1, B_{i-1})$. Such an adversary model naturally captures many applications, in which the data structure is used in an *iterative process*, and the input to the data structure depends on the output from last iteration.

We will now describe how our data structure can be extended to handle an adaptive adversary. We store all initial factor matrices $A_1, \ldots, A_q$. During an update, suppose we are given a pair $(i, B_i)$. Instead of using the sketching matrices correspond to the $i$-th leaf during initialization, we instead choose fresh sketching matrices along the path from the leaf to the root. Since the only randomness being used is the sketching matrices along the path, by using fresh randomness, our data structure works against adaptive adversary.

Such modification does not affect the update time too much: during each update, we need to apply the newly-generated base sketch $C_i^{\text{new}}$ to both $A_i$ and $B$, incurring a time of $\widetilde{O}(\text{nnz}(A_i) + \text{nnz}(B))$. Along the path, the data structure repeatedly apply $T_{k,\ell}^{\text{new}}$ to its two children, which takes $\widetilde{O}(md)$ time. The overall time is thus

$$\widetilde{O}(\text{nnz}(A_i) + \text{nnz}(B) + md).$$

Thus, in later applications of our data structure, we present the runtime without this modification.

# 6 Faster Dynamic Tensor Product Algorithms with Dynamic Tree

In this section, we instantiate the dynamic tree data structure introduced in section 5 for three applications: dynamic tensor product regression, dynamic tensor spline regression, and dynamic tensor low rank approximation.

## 6.1 Dynamic Tensor Product Regression

We present an algorithm (Algorithm. 2) for solving the dynamic Tensor product regression problem. Our algorithm uses the dynamic tree data structure to maintain a sketch of the design matrix and updates it efficiently. Additionally, we maintain a sketch of the label vector $b$ via $\Pi^q b$. During the query phase, we output the solution using the sketch of the tensor product design matrix and the sketch of the label vector. We have the following guarantees, the proof of which we defer to Appendix C:

**Theorem 6.1** (Informal version of Theorem C.1). *There is an algorithm (Algorithm 2) for dynamic Tensor product regression problem with the following procedures:*

- INITIALIZE($A_1 \in \mathbb{R}^{n_1 \times d_1}, A_2 \in \mathbb{R}^{n_2 \times d_2}, \ldots, A_q \in \mathbb{R}^{n_q \times d_q}, m \in \mathbb{N}_+, C_{\text{base}}, T_{\text{base}}, b \in \mathbb{R}^{n_1 \cdots n_q}$): *Given matrices $A_1 \in \mathbb{R}^{n_1 \times d_1}, A_2 \in \mathbb{R}^{n_2 \times d_2}, \ldots, A_q \in \mathbb{R}^{n_q \times d_q}$,, a sketching dimension $m$, families of base sketches $C_{\text{base}}, T_{\text{base}}$ and a label vector $b \in \mathbb{R}^{n_1 \cdots n_q}$, the data-structure pre-processes in time $\widetilde{O}(\sum_{i=1}^q \mathrm{nnz}(A_i) + qmd + m \cdot \mathrm{nnz}(b))$.*

- UPDATE($i \in [q], B \in \mathbb{R}^{n_i \times d_i}$): *Given a matrix $B$ and an index $i \in [q]$, the data structure updates the approximation to $A_1 \otimes \ldots \otimes (A_i + B) \otimes \ldots \otimes A_q$ in time $\widetilde{O}(\mathrm{nnz}(B) + md)$.*

- QUERY: *Query outputs an approximate solution $\widehat{x}$ to the Tensor product regression problem where $\| \bigotimes_{i=1}^q A_i \widehat{x} - b \|_2 = (1 \pm \varepsilon) \| \bigotimes_{i=1}^q A_i x^* - b \|_2$ with probability at least $1 - \delta$ in time $O(md^{\omega - 1} + d^\omega)$ where $x^*$ is an optimal solution to the Tensor product regression problem i.e. $x^* = \arg\min_{x \in \mathbb{R}^d} \| \bigotimes_{i=1}^q A_i x - b \|_2$.*

To realize the input sparsity time initialize and update time, we can use the combination of OSNAP and TENSORSRHT, which gives a sketching dimension

$$m = \varepsilon^{-2} q d^2 \log(1/\delta).$$

Hence, the update time is $\widetilde{O}(\mathrm{nnz}(B) + \varepsilon^{-2} q d^3 \log(1/\delta))$. Compared to the static algorithm [DJS$^+$19], we note that their algorithm is suitable to be modified into dynamic setting, with a running time of $\widetilde{O}(\mathrm{nnz}(A_i) + \mathrm{nnz}(B) + \varepsilon^{-2}(q + d)d/\delta))$, their algorithm cannot accommodate general dynamic setting due to the $1/\delta$ dependence in running time. Suppose the length dynamic sequence is at least $d$, then one requires the failure probability to be at most $\frac{1}{d}$ for union bound. In such scenario, the algorithm of [DJS$^+$19] is strictly worse. [FFG22] improves the dependence on $d$, yielding a subquadratic update time in the form of $\widetilde{O}((\mathrm{nnz}(A_i) + \mathrm{nnz}(B) + \varepsilon^{-2} q^2 d_i^\omega + \varepsilon^{-1} d^{2-\theta}) \log(1/\delta))$. While their algorithm is more efficient in solving the regression problem, their approach does not scale with the statistical dimension of the problem, and is geared only towards regression, whereas our data structure also handles low rank approximation.

## 6.2 Dynamic Tensor Spline Regression

Spline regression models are a well studied class of regression models [MC01]. In particular, Spline regression problems involving tensor products arise in the context of B-Splines and P-Splines [EM96]. For a very brief but relevant introduction to Spline regression, we refer the reader to [DSSW18]. In this section, we first define the Spline regression problem as it pertains to our sketching application. Algorithm 3, which shows how we can use our DYNAMICTENSORTREE data structure to solve a dynamic version of the Spline regression problem is provided in Appendix D.

**Definition 6.2** (Spline Regression). In the Spline regression problem, given a design matrix $A \in \mathbb{R}^{m \times n}$, a target vector $x \in \mathbb{R}^n$, a regularization matrix $L \in \mathbb{R}^{p \times d}$, and a non-negative penalty coefficient $\lambda \in \mathbb{R}$, the goal is to optimize

$$\min_{x \in \mathbb{R}^n} \|Ax - b\|_2^2 + \lambda \cdot \|Lx\|_2^2 \tag{1}$$

Notice that the classic ridge regression problem is a special case of spline regression (when $L = I$). In the TENSOR SPLINE REGRESSION problem, we want to solve the Spline regression problem where the design matrix $A$ can be decomposed as the tensor product of multiple smaller matrices i.e. $A = A_1 \otimes \cdots \otimes A_q$. Our DYNAMICTENSORTREE data structure can be used to solve a dynamic version of TENSOR SPLINE REGRESSION. In our dynamic model, at each time step, the update is of the form $i \in [q], B \in \mathbb{R}^{n_i \times d_i}$, and the data structure has to update $A_i \leftarrow A_i + B$.

Before introducing our main result, we define an important metric to measure the rank of a Spline regression:

**Definition 6.3** (Statistical dimension of Splines). For matrix $A \in \mathbb{R}^{n \times d}$ and $L \in \mathbb{R}^{p \times d}$ with $\text{rank}(L) = p$ and $\text{rank}\left(\begin{bmatrix} A \\ L \end{bmatrix}\right) = d$. Given the generalized singular value decomposition [GVL13] of $(A, L)$ such that $A = U \begin{bmatrix} \Sigma & 0_{p \times (n-p)} \\ 0_{(n-p) \times p} & I_{d-p} \end{bmatrix} RQ^\top, L = V \begin{bmatrix} \Omega & 0_{p \times (n-p)} \end{bmatrix} RQ^\top$, where $\Sigma = \text{diag}(\sigma_1, \ldots, \sigma_p), \Omega = \text{diag}(\mu_1, \ldots, \mu_p) \in \mathbb{R}^{p \times p}$ are diagonal matrices. Finally, let $\gamma_i = \sigma_i / \mu_i$ for $i \in [p]$.

The statistical dimension of the Spline is defined as $\text{sd}_\lambda(A, L) = \sum_{i=1}^{p} 1/(1 + \lambda/\gamma_i^2) + d - p$.

The statistical dimension of the Spline can be much smaller than $d$, and we will show that the sketching dimension depends on the statistical dimension instead of $d$.

Let $x^*$ denote an optimal solution of the Spline regression problem, i.e.
$$x^* = \arg\min_{x \in \mathbb{R}^n} \|Ax - b\|_2^2 + \lambda \cdot \|Lx\|_2^2$$
and let $\text{OPT} := \|Ax^* - b\|_2^2 + \lambda \cdot \|Lx^*\|_2^2$. We can compute $x^*$ given $A, b, L, \lambda$ by $x^* = (A^\top A + \lambda L^\top L)^{-1} A^\top b$.

**Theorem 6.4** (Guarantees for DTSREGRESSION. Informal version of Theorem D.5). *There exists an algorithm (Algorithm 3) with the following procedures:*

- INITIALIZE($A_1 \in \mathbb{R}^{n_1 \times d_1}, \ldots, A_q \in \mathbb{R}^{n_q \times d_q}$): *Given matrices $A_1 \in \mathbb{R}^{n_1 \times d_1}, \ldots, A_q \in \mathbb{R}^{n_q \times d_q}$, the data structure pre-processes in time*
$$\widetilde{O}(\sum_{i=1}^{q} \text{nnz}(A_i) + qmd + m \cdot \text{nnz}(b)).$$

- UPDATE($i \in [q], B \in \mathbb{R}^{n_i \times d_i}$): *Given an index $i \in [q]$ and an update matrix $B \in \mathbb{R}^{n \times d}$, the data structure updates in time $\widetilde{O}(\text{nnz}(B) + md)$.*

- QUERY: *Let $A$ denote the tensor product $\bigotimes_{i=1}^{q} A_i$. Query outputs a solution $\widetilde{x} \in \mathbb{R}^n$ with constant probability such the $\widetilde{x}$ is an $\varepsilon$ approximate to the Tensor Spline Regression problem i.e.*
$$\|A\widetilde{x} - b\|_2^2 + \lambda \cdot \|L\widetilde{x}\|_2^2 \le (1 + \varepsilon) \cdot \text{OPT}$$
*where $\text{OPT} = \min_{x \in \mathbb{R}^n} \|Ax - b\|_2^2 + \lambda \cdot \|Lx\|_2^2$. Query takes time $md^{(\omega-1)} + pd^{(\omega-1)} + d^\omega$.*

We defer the proof of the above theorem to Appendix D.

The sketching dimension $m$ here depends on the statistical dimension of the Spline, which is at most $d$. To realize the above runtime, one can pick OSNAP and TENSORSHRT, which gives $m = \varepsilon^{-1} q \, \text{sd}_\lambda^2(A, L) \log(1/\delta)$ and an update time $\widetilde{O}(\text{nnz}(B) + \varepsilon^{-1} q \, \text{sd}_\lambda^2(A, L) d \log(1/\delta))$. The improved dependence on $\varepsilon^{-1}$ comes from using approximate matrix product directly, instead of OSE.

## 6.3 Dynamic Tensor Low Rank Approximation

In this section, we consider a problem studied in [DJS+19]: given matrices $A_1, \ldots, A_q$, the goal is to compute a low rank approximation for the tensor product matrix $A_1 \otimes \ldots \otimes A_q \in \mathbb{R}^{n \times d}$, specifically, a rank-$k$ approximation $B \in \mathbb{R}^{n \times d}$ such that
$$\|B - A\|_F \le (1 + \varepsilon) \text{OPT}_k,$$

where $\text{OPT}_k = \min_{\text{rank-}k\ A'} \|A' - A\|_F$ and $A = \bigotimes_{i=1}^{q} A_i$. The [DJS$^+$19] algorithm works as follows: they pick $q$ independent COUNTSKETCH matrices [CCFC02] with $O(\varepsilon^{-2}qk^2)$ rows, then apply each sketch matrix $S_i$ to $A_i$. After that, they form the tensor product $M = \bigotimes_{i=1}^{q} S_i A_i$ and run SVD on $M = U\Sigma V^\top$. Finally, they output $B = AU_k^\top U_k$ in factored form (as matrices $A_1, \ldots, A_q, U_k$), where $U_k \in \mathbb{R}^{k \times d}$ denotes the top $k$ right singular vectors. By doing so, they achieve an overall running time of

$$O(\sum_{i=1}^{q} \text{nnz}(A_i) + \varepsilon^{-2q}q^q k^{2q} d^{\omega-1})$$

Two central issues around their algorithm are 1). exponential dependence on parameters $\varepsilon^{-1}, q, k$ and 2). the algorithm is inherently static, so a natural way to make it dynamic is to apply $S_i$ to the update $B$, reforming the tensor product of sketches and compute the SVD.

We address these two issues simultaneously by using our tree data structure. For the sketching dimension, we observe it is enough to design a sketch that is $(\varepsilon, \delta, k, d, n)$-OSE and $(\varepsilon/k, \delta)$-approximate matrix product, therefore, we can set sketching dimension $m = \varepsilon^{-2}qk^2 \log(1/\delta)$. As we will show later, this gives a much improved running time. By using the dynamic tree, we also provide a dynamic algorithm.

**Theorem 6.5** (Guarantees for LOWRANKMAINTENANCE. Informal version of Theorem E.3)**.**
*There exists an algorithm (Algorithm 4) that has the following procedures*

- INITIALIZE($A_1 \in \mathbb{R}^{n_1 \times d_1}, \ldots, A_q \in \mathbb{R}^{n_q \times d_q}$)*: Given matrices $A_1 \in \mathbb{R}^{n_1 \times d}, \ldots, A_q \in \mathbb{R}^{n_q \times d_q}$, the data structure processes in time*

$$\widetilde{O}(\sum_{i=1}^{q} \text{nnz}(A_i) + qmd).$$

- UPDATE($i \in [q], B \in \mathbb{R}^{n_i \times d_i}$)*: Given an index $i \in [q]$ and an update matrix $B \in \mathbb{R}^{n_i \times d_i}$, the data structure updates the approximation for $A_1 \otimes \ldots \otimes (A_i + B) \otimes \ldots \otimes A_q$ in time*
$$\widetilde{O}(\text{nnz}(B) + md).$$

- QUERY*: Let $A$ denote the tensor product $\bigotimes_{i=1}^{q} A_i$. The data structure outputs a rank-$k$ approximation $C$ such that*
$$\|C - A\|_F \le (1 + \varepsilon) \min_{\text{rank-}k\ A'} \|A' - A\|_F.$$
*The time to output $C$ is $\widetilde{O}(md^{\omega-1})$.*

By choosing $m = \varepsilon^{-2}qk^2 \log(1/\delta)$, we can provide good guarantees for low rank approximation. This gives

- Initialization time in $\widetilde{O}(\sum_{i=1}^{q} \text{nnz}(A_i) + \varepsilon^{-2}qdk^2 \log(1/\delta))$;
- Update time in $\widetilde{O}(\text{nnz}(B) + \varepsilon^{-2}qdk^2 \log(1/\delta))$;
- Query time in $\widetilde{O}(\varepsilon^{-2}qd^{\omega-1}k^2 \log(1/\delta))$.

Without using fast matrix multiplication, our algorithm improves upon the prior state-of-the-art [DJS$^+$19] on all parameters even in the static setting.

## 7  Conclusion & Future Directions

In this work, we initiate the study of the Dynamic Tensor Regression, Dynamic Tensor Spline Regression, and Dynamic Tensor Low-Rank Approximation problems, and design a tree data structure DYNAMICTENSORTREE which can be used to provide fast dynamic algorithms for these problems. The problems we study are very interesting and there are a lot of exciting future directions, some of which we discuss below. First, it is unclear whether the algorithms we proposed are optimal and so having lower bounds for the update times would be an important direction. Second, we have focused exclusively on regression problems with the $\ell_2$ norm whereas solving Dynamic Tensor Regression and Dynamic Tensor Spline Regression with respect to the more robust $\ell_1$-norm as well as for more general $p$-norms are also very appealing.

## Acknowledgments

We would like to thank all the NeurIPS 2022 reviewers of our paper and also the ICML 2022 reviewers who reviewed an earlier version of this work. Most of this work was done while LZ was at CMU. AR was supported in-part by NSF grants CCF-1652491, CCF-1934931, and CCF-1955351 during the preparation of this manuscript.

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
