# Appendix

**Roadmap.**    In this appendix, we provide omitted details and proofs for our main paper. In particular, section A of our appendix corresponds to section 3 of our main paper, section B to section 5, section C to section 6.1, section D to section 6.2, and section E to section 6.3.

## A    Background

In this section, we provide definitions for some terms used in the main paper and rest of the appendix.

**Definition A.1** (Statistical Dimension). For any $\lambda \geq 0$ and any positive semidefinite matrix $M \in \mathbb{R}^{n \times n}$, the $\lambda$-statistical dimension of $M$ is

$$\mathrm{sd}_\lambda(M) := \mathrm{tr}[M(M + \lambda I_n)^{-1}]$$

**Definition A.2** (Oblivious Subspace Embedding [Sar06, AKK$^+$20]). Let $\varepsilon, \delta \in (0, 1), \mu > 0$, and $d, n \geq 1$ be integers. An $(\varepsilon, \delta, \mu, d, n)$-Oblivious Subspace Embedding (OSE) is a distribution over $m \times d$ matrices with the guarantee that for any fixed matrix $A \in \mathbb{R}^{d \times n}$ with $\mathrm{sd}_\lambda(A^\top A) \leq \mu$, we have for any vector $x \in \mathbb{R}^n$, with probability at least $1 - \delta$,

$$\|\Pi A x\|_2 + \lambda \|x\|_2 = (1 \pm \varepsilon)(\|Ax\|_2 + \lambda \|x\|_2).$$

**Definition A.3** (Approximate Matrix Product). Given $\varepsilon, \delta > 0$, we say that a distribution $\mathcal{D}$ over $m \times d$ matrices $\Pi$ has the $(\varepsilon, \delta)$-approximate matrix product property if for every $C, D \in \mathbb{R}^{d \times n}$, with probability at least $1 - \delta$,

$$\|C^\top \Pi^\top \Pi D - C^\top D\|_F \leq \varepsilon \|C\|_F \|D\|_F.$$

**Definition A.4** (OSNAP matrix [NN13]). For every sparsity parameter $s$, target dimension $m$, and positive integer $d$, the OSNAP transform with sparsity $s$ is defined as

$$S_{r,j} = \frac{1}{\sqrt{s}} \cdot \delta_{r,j} \cdot \sigma_{r,j},$$

for all $r \in [m]$, $j \in [d]$, where $\sigma_{r,j} \in \{\pm 1\}$ are independent and uniform Rademacher random variables and $\delta_{r,j}$ are Bernoulli random variables satisfying

- For every $i \in [d]$, $\sum_{r \in [m]} \delta_{r,i} = s$. In other words, every column has exactly $s$ nonzero entries.

- For all $r \in [m]$ and all $i \in [d]$, $\mathbb{E}[\delta_{r,i}] = s/m$.

- The $\delta_{r,i}$'s are negatively correlated: $\forall T \subset [m] \times [d]$, $\mathbb{E}[\prod_{(r,i) \in T} \delta_{r,i}] \leq \prod_{(r,i) \in T} \mathbb{E}[\delta_{r,i}] = \left(\frac{s}{m}\right)^{|T|}$.

**Definition A.5** (TENSORSRHT [AKK$^+$20]). We define the TENSORSRHT $S : \mathbb{R}^d \times \mathbb{R}^d \to \mathbb{R}^m$ as $S = \frac{1}{\sqrt{m}} P \cdot (HD_1 \times HD_2)$, where each row of $P \in \{0, 1\}^{m \times d}$ contains only one 1 at a random coordinate, one can view $P$ as a sampling matrix. $H$ is a $d \times d$ Hadamard matrix, and $D_1, D_2$ are two $d \times d$ independent diagonal matrices with diagonals that are each independently set to be a Rademacher random variable (uniform in $\{-1, 1\}$).

**Definition A.6** (COUNTSKETCH matrix [CCFC02]). Let $h : [n] \to [b]$ be a random 2-wise independent hash function and $\sigma : [n] \to \{-1, +1\}$ be a random 4-wise independent hash function. Then $R \in \mathbb{R}^{b \times n}$ is a COUNTSKETCH matrix if we set $R_{h(i),i} = \sigma(i)$ for all $i \in [n]$ and all the other entries of $R$ to zero.

**Definition A.7** (TENSORSKETCH matrix [Pag13, PP13]). Let $h_1, h_2 : [m] \to [s]$ be 3-wise independent hash functions, also let $\sigma : [m] \to \{\pm 1\}$ be a 4-wise independent random sign function. The degree two TENSORSKETCH transform, $S : \mathbb{R}^m \times \mathbb{R}^m \to \mathbb{R}^s$ is defined as follows: for any $i, j \in [m]$ and $r \in [s]$,

$$S_{r,(i,j)} = \sigma_1(i) \cdot \sigma_2(j) \cdot \mathbf{1}[h_1(i) + h_2(j) = r \bmod s].$$

**Definition A.8** (Subsampled Randomized Hadamard Transform (SRHT) matrix [LDFU13]). We say $R \in \mathbb{R}^{b \times n}$ is a subsampled randomized Hadamard transform matrix[4] if it is of the form $R = \sqrt{n/b} \cdot SHD$, where $S \in \mathbb{R}^{b \times n}$ is a random matrix whose rows are $b$ uniform samples (without replacement) from the standard basis of $\mathbb{R}^n$, $H \in \mathbb{R}^{n \times n}$ is a normalized Walsh-Hadamard matrix, and $D \in \mathbb{R}^{n \times n}$ is a diagonal matrix whose diagonal elements are i.i.d. Rademacher random variables.

# B    Proofs for Dynamic Tree Data Structure (section 5)

In this section, we provide proofs for the running time and approximation guarantees of our dynamic tree data structure (Algorithm 1).

## B.1    Running Time of Algorithm 1

**Theorem B.1** (Formal version of Theorem 5.1). *There is an algorithm (Algorithm 1) that has the following procedures:*

- INITIALIZE*($A_1 \in \mathbb{R}^{n_1 \times d_1}, A_2 \in \mathbb{R}^{n_2 \times d_2}, \ldots, A_q \in \mathbb{R}^{n_q \times d_q}, m \in \mathbb{N}_+, C_{\mathrm{base}}, T_{\mathrm{base}}$): Given matrices $A_1 \in \mathbb{R}^{n_1 \times d_1}, A_2 \in \mathbb{R}^{n_2 \times d_2}, \ldots, A_q \in \mathbb{R}^{n_q \times d_q}$, a sketching dimension $m$, families of base sketches $C_{\mathrm{base}}$ and $T_{\mathrm{base}}$, the data-structure pre-processes in time $\widetilde{O}(\sum_{i=1}^{q} \mathrm{nnz}(A_i) + qmd)$ where $d = d_1 d_2 \ldots d_q$.*

- UPDATE*($i \in [q], B \in \mathbb{R}^{n_i \times d_i}$): Given a matrix $B$ and an index $i \in [q]$, the data structure updates the approximation to $A_1 \otimes \ldots \otimes (A_i + B) \otimes \ldots \otimes A_q$ in time $\widetilde{O}(\mathrm{nnz}(B) + md)$.*

Note: We will assume that $n_1 = n_2 = \cdots = n_q$ and $d_1 = d_2 = \cdots = d_q$ for the following computations. To modify our DYNAMICTENSORTREE to work for the non-uniform case, we only need to choose the base sketches at the leaves of the appropriate dimension and also have different dimensions for the internal nodes.

*Proof.* **Proof of INITIALIZATION.** The initialization part has 2 key steps - applying the $T_{\mathrm{base}}$ sketches to the leaves and applying $S_{\mathrm{base}}$ sketches to each of the internal nodes.

- Applying a COUNTSKETCH matrix $C_i : \mathbb{R}^{m \times n_i}$ to any matrix $A_i : \mathbb{R}^{n_i \times d_i}$ takes input sparsity time $O(\mathrm{nnz}(A_i))$. Therefore, the total time for this step is $O(\sum_{i=1}^{q} \mathrm{nnz}(A_i))$. If OSNAP is used, this is true up to polylogarithmic factors.

- For all internal nodes, let us consider the time of computing one node at level $\ell \in [\log_2 q]$ using TENSORSRHT. Consider the computation of $J_{k,\ell} \in \mathbb{R}^{m \times d_1^{2^\ell}}$, we pick a TEN-SORSRHT $T_{k,\ell} \in \mathbb{R}^{m \times m^2}$ and apply it to $J_{2k,\ell-1} \otimes J_{2k+1,\ell-1}$. This can be viewed as computing the tensor product of column vectors of $J_{2k,\ell-1}$ and $J_{2k+1,\ell-1}$: each column of $J_{k,\ell}$ is a tensor product of two columns of its two children after applying $T_{k,\ell}$. Each column takes time $O(m \log m)$ and the total number of columns of $J_{k,\ell}$ is $d_1^{2^\ell}$. Hence, the time of forming $J_{k,\ell}$ is $O(md_1^{2^\ell} \log m)$.

- For each level, there are $q/2^\ell$ nodes, so $O(q/2^\ell m d_1^{2^\ell})$ time in total for level $\ell$.

- Sum over all levels for $\ell \in [\log q]$, we have

$$m \log m \left( \sum_{\ell=1}^{\log q} \frac{q}{2^\ell} d_1^{2^\ell} \right) \leq m \log m \left( \sum_{\ell=1}^{\log q} \frac{q}{2^\ell} d_1^q \right)$$

$$= md \log m \left( \sum_{\ell=1}^{\log q} \frac{q}{2^\ell} \right)$$

$$\leq 2qmd \cdot \log m.$$

---

[4] In this case, we require $\log n$ to be an integer.

- Thus, the total time for initialization is

$$\widetilde{O}(\sum_{i=1}^{q} \mathrm{nnz}(A_i) + qmd).$$

**Proof of UPDATE.** We note that UPDATE can be decomposed into the following parts:

- Apply COUNTSKETCH $C_i$ to the update matrix $B$, which takes $O(\mathrm{nnz}(B))$ time. If OS-NAP or SRHT are used, then this is true up to polylogarithmic factors.

- Updating the leaf node takes $O(md_i)$ time because $C_i B \in \mathbb{R}^{m \times d_i}$.

- For each level $\ell \in [\log_2 q]$, we first compute the update tree by applying TENSORSKETCH $T_{k,\ell} \in \mathbb{R}^{m \times m^2}$ to $J_{2k,\ell-1} \otimes J_{2k+1,\ell-1}$ which takes time $O(md_1^{2^\ell} \log m)$.

- We also need to update **one** internal node for each level, which takes $\widetilde{O}(md_1^{2^\ell})$ time. This is at most $O(md \log m)$ for the root. Since there are $\log q$ levels, the total time for updating all internal nodes is also $\widetilde{O}(md)$.

Hence, the total runtime of UPDATE is

$$\widetilde{O}(\mathrm{nnz}(B) + md_1 + md)$$
$$= \widetilde{O}(\mathrm{nnz}(B) + md).$$

$\square$

### B.2 Approximation Guarantee of Algorithm 1

In this section, we provide the approximation guarantee of Algorithm 1.

**Theorem B.2.** *Let $A_1 \in \mathbb{R}^{n_1 \times d_1}, \ldots, A_q \in \mathbb{R}^{n_q \times d_q}$ be the input matrices, $m \in \mathbb{N}_+$ be the sketching dimension, let $C_{\mathrm{base}}, T_{\mathrm{base}}$ be appropriate family of sketching matrices. Let $\Pi^q$ be the sketching matrix generated by Algorithm 1. Then for any $x \in \mathbb{R}^d$, we have that with probability at least $1 - \delta$,*

$$\|\Pi^q \bigotimes_{i=1}^{q} A_i x\|_2 = (1 \pm \varepsilon)\|\bigotimes_{i=1}^{q} A_i x\|_2,$$

*if*

- $C_{\mathrm{base}}$ *is* COUNTSKETCH, $T_{\mathrm{base}}$ *is* TENSORSKETCH *and* $m = \Omega(\varepsilon^{-2} q d^2 1/\delta)$.

- $C_{\mathrm{base}}$ *is* OSNAP, $T_{\mathrm{base}}$ *is* TENSORSRHT *and* $m = \widetilde{\Omega}(\varepsilon^{-2} q^4 d \log(1/\delta))$.

- $C_{\mathrm{base}}$ *is* OSNAP, $T_{\mathrm{base}}$ *is* TENSORSRHT *and* $m = \widetilde{\Omega}(\varepsilon^{-2} q d^2 \log(1/\delta))$.

First we will discuss the theorems of [AKK$^+$20] which lead to our approximation guarantees and then discuss the specifics for each choice of base sketches seperately.

**Lemma B.3** (Theorem 1 of [AKK$^+$20]). *For every positive integers $n, q, d$, every $\varepsilon, \mathtt{sd}_\lambda > 0$, there exists a distribution on linear sketches $\Pi^q \in \mathbb{R}^{m \times d^q}$ such that: 1). If $m = \Omega(\varepsilon^{-2} q \, \mathtt{sd}_\lambda^2)$, then $\Pi^q$ is an $(\varepsilon, 1/10, \mathtt{sd}_\lambda, d^q, n)$-OSE (Def. A.2). 2). If $m = \Omega(\varepsilon^{-2} q)$, then $\Pi^q$ has the $(\varepsilon, 1/10)$-approximate matrix product (Def. A.3).*

**Lemma B.4** (Theorem 2 of [AKK$^+$20]). *For every positive integers $n, q, d$, every $\varepsilon, \mathtt{sd}_\lambda > 0$, there exists a distribution on linear sketches $\Pi^q \in \mathbb{R}^{m \times d^q}$ such that: 1). If $m = \widetilde{\Omega}(\varepsilon^{-2} q \, \mathtt{sd}_\lambda^2)$, then $\Pi^q$ is an $(\varepsilon, 1/\mathrm{poly}(n), \mathtt{sd}_\lambda, d^q, n)$-OSE (Def. A.2). 2). If $m = \widetilde{\Omega}(\varepsilon^{-2} q)$, then $\Pi^q$ has the $(\varepsilon, 1/\mathrm{poly}(n))$-approximate matrix product (Def. A.3).*

**Lemma B.5** (Theorem 3 of [AKK$^+$20]). *For every positive integers $n, q, d$, every $\varepsilon, \mathtt{sd}_\lambda > 0$, there exists a distribution on linear sketches $\Pi^q \in \mathbb{R}^{m \times d^q}$ which is an $(\varepsilon, 1/\mathrm{poly}(n), \mathtt{sd}_\lambda, d^q, n)$-OSE (Def. A.2), provided that the integer $m$ satisfies $m = \widetilde{\Omega}(q^4 \, \mathtt{sd}_\lambda / \varepsilon^2)$*

We note that the 3 results due to [AKK$^+$20] are actually stronger than what we need here, since the statements say that they are subspace embedding for *all* matrices of proper size. Also, when the statistical dimension $\mathrm{sd}_\lambda$ is smaller than the rank of the design matrix, the sketching dimension is even smaller, which is important in our Spline regression application.

We then proceed to prove the individual guarantees for different choice of base sketches item by item.

- By Lemma B.3, we know that in order for $\Pi^q$ to be an OSE, we will require $m = \Omega(\varepsilon^{-2}qd^2 1/\delta)$. Note that the statistical dimension $\mathrm{sd}_\lambda$ is $d$ in this case. We point out that the dependence on success probability is $1/\delta$, this means that we can not obtain a high probability version. This is the main reason that Lemma B.3 only provides an OSE with constant probability.

- By Lemma B.5, we need $m = \widetilde{\Omega}(\varepsilon^{-2}q^4 d\log(1/\delta))$. We get a nearly linear dependence on $d$ in the expense of worse dependence on $q$.

- By Lemma B.4, we will require $m = \widetilde{\Omega}(\varepsilon^{-2}qd^2\log(1/\delta))$. In this scenario, the dependence on $\delta$ is $\log(1/\delta)$, hence we can aim for a high probability guarantee for our OSE.

## C  Proofs for Dynamic Tensor Product Regression (section 6.1)

---

**Algorithm 2** Dynamic Tensor Product Regression

---

1: **data structure** DTREGRESSION
2:
3: **members**
4:     DYNAMICTENSORTREE DTT
5:     $A_1 \in \mathbb{R}^{n_1 \times d_1}, \ldots, A_q \in \mathbb{R}^{n_q \times d_q}$
6:     $\widetilde{b} \in \mathbb{R}^m$
7: **end members**
8:
9: **procedure** INITIALIZE($A_1 \in \mathbb{R}^{n_1 \times d_1}, \ldots, A_q \in \mathbb{R}^{n_q \times d_q}, m, C_{\text{base}}, T_{\text{base}}, b \in \mathbb{R}^n$)
10:     DTT.INITIALIZE($A_1, \ldots, A_q, m, C_{\text{base}}, T_{\text{base}}$)
11:     $A_1 \leftarrow A_1, \ldots, A_q \leftarrow A_q$
12:     Let $\Pi_q \in \mathbb{R}^{m \times n}$ be the sketching matrix corresponds to DTT
13:     $\widetilde{b} \leftarrow \Pi_q b$
14: **end procedure**
15:
16: **procedure** QUERY()
17:     $M \leftarrow$ DTT.$J_{0,0}$
18:     $\widetilde{x} \leftarrow \arg\min_{x \in \mathbb{R}^n} \|Mx - \widetilde{b}\|_2^2$
19:     **return** $x$
20: **end procedure**

---

**Theorem C.1** (Formal version of Theorem 6.1). *There is a dynamic data structure for dynamic Tensor product regression problem with the following procedures:*

- INITIALIZE*($A_1 \in \mathbb{R}^{n_1 \times d_1}, A_2 \in \mathbb{R}^{n_2 \times d_2}, \ldots, A_q \in \mathbb{R}^{n_q \times d_q}, m \in \mathbb{N}_+, C_{\text{base}}, T_{\text{base}}, b \in \mathbb{R}^{n_1 \cdots n_q}$): Given matrices $A_1 \in \mathbb{R}^{n_1 \times d_1}, A_2 \in \mathbb{R}^{n_2 \times d_2}, \ldots, A_q \in \mathbb{R}^{n_q \times d_q}$,, a sketching dimension $m$, families of base sketches $C_{\text{base}}, T_{\text{base}}$ and a label vector $b \in \mathbb{R}^{n_1 \cdots n_q}$, the data-structure pre-processes in time $\widetilde{O}(\sum_{i=1}^q \mathrm{nnz}(A_i) + qmd + m \cdot \mathrm{nnz}(b))$.*

- UPDATE*($i \in [q], B \in \mathbb{R}^{n_i \times d_i}$): Given a matrix $B$ and an index $i \in [q]$, the data structure updates the approximation to $A_1 \otimes \ldots \otimes (A_i + B) \otimes \ldots \otimes A_q$ in time $\widetilde{O}(\mathrm{nnz}(B) + md)$.*

- QUERY*: Query outputs an approximate solution $\widehat{x}$ to the Tensor product regression problem where $\|\bigotimes_{i=1}^q A_i \widehat{x} - b\|_2 = (1 \pm \varepsilon)\|\bigotimes_{i=1}^q A_i x^* - b\|_2$ with probability at least $1 - \delta$ in time $O(md^{\omega-1} + d^\omega)$ where $x^*$ is an optimal solution to the Tensor product regression problem i.e. $x^* = \arg\min_{x \in \mathbb{R}^d} \|\bigotimes_{i=1}^q A_i x - b\|_2$.*

*Proof.* **Proof of INITIALIZATION**. The initialization part has 3 steps:

- Initializing DTT takes time $\widetilde{O}(\sum_{i=1}^{q} \text{nnz}(A_i) + qmd)$ from Theorem B.1.

- Initializing all the matrices $A_i, i \in [q]$ takes time $\widetilde{O}(\sum_{i=1}^{q} \text{nnz}(A_i))$.

- Computing $\widehat{b}$ takes time $\widetilde{O}(m \cdot \text{nnz}(b))$.

Therefore, the total time for initialization is

$$\widetilde{O}(\sum_{i=1}^{q} \text{nnz}(A_i) + qmd + \sum_{i=1}^{q} \text{nnz}(A_i) + m \cdot \text{nnz}(b))$$
$$= \widetilde{O}(\sum_{i=1}^{q} \text{nnz}(A_i) + qmd + m \cdot \text{nnz}(b))$$

**Proof of UPDATE**. We just update our DTT data structure. This takes time $\widetilde{O}(\text{nnz}(B) + md^{3/2})$ from Theorem B.1.

**Proof of QUERY time**. We can view the Query part as having the following 5 steps:

- Initializing $M$ takes time $\widetilde{O}(md)$ since $M \in \mathbb{R}^{m \times d}$.

- $M^\top(\widetilde{b})$, this takes $O(md)$ time because we are multiplying a $d \times m$ matrix with a $m \times 1$ vector.

- $M^\top \cdot M$, this step takes $\mathcal{T}_{\text{mat}}(d, m, d)$ time as we are multiplying an $d \times m$ matrix with an $m \times d$ matrix.

- $(M^\top M)^{-1}$ this step takes $\mathcal{T}_{\text{mat}}(d, d, d)$ time as it is inverting $M^\top M$ which is of size $d \times d$.

- $(M^\top M)^{-1}(M^\top \widetilde{b})$. In this step, we are multiplying $(M^\top M)^{-1}$ which is a $d \times d$ matrix with a vector $M^\top \widetilde{b}$ of size $d$. This takes time $O(d^2)$.

Overall, the total running time of QUERY is

$$\widetilde{O}(md + md + \mathcal{T}_{\text{mat}}(d, m, d) + \mathcal{T}_{\text{mat}}(d, d, d) + d^2)$$
$$= \widetilde{O}(md + md^{(\omega-1)} + d^\omega + d^2)$$
$$= \widetilde{O}(md^{(\omega-1)} + d^\omega)$$

**Proof of Correctness for QUERY**. The proof of correctness for Query follows from Theorem B.2 identically to the proof of Theorem 3.1 in [DSSW18]. $\qquad\square$

## D Omitted details and proofs for Dynamic Tensor Spline Regression (section 6.2)

To specify the sketching dimension $m$ needed , we use the following definition of *Statistical Dimension* for Splines [DSSW18]:

**Definition D.1** (Statistical Dimension for Splines). For the Spline regression problem specified by Eq. 1, the statistical dimension is defined as $\text{sd}_\lambda(A, L) = \sum_i 1/(1+\lambda/\gamma_i^2)+d-p$. Here $\{\gamma_i, i \in [p]\}$ are the *generalized singular values* of $(A, L)$ which are defined as follows.

**Definition D.2** (Generalized singular values). For matrices $A \in \mathbb{R}^{n \times d}$ and $L \in \mathbb{R}^{p \times d}$ such that $\text{rank}(L) = p$ and $\text{rank}\left(\begin{bmatrix} A \\ L \end{bmatrix}\right) = d$, the generalized singular value decomposition (GSVD) [GVL13] of $(A, L)$ is given by the pair of factorizations $A =$

$U \begin{bmatrix} \Sigma & 0_{p \times (d-p)} \\ 0_{(d-p) \times p} & I_{d-p} \end{bmatrix} RQ^\top$ and $L = V \begin{bmatrix} \Omega & 0_{p \times (d-p)} \end{bmatrix} RQ^\top$, where $U \in \mathbb{R}^{n \times d}$ has orthonormal columns, $V \in \mathbb{R}^{p \times p}$, $Q \in \mathbb{R}^{d \times d}$ are orthogonal, $R \in \mathbb{R}^{d \times d}$ is upper triangular and nonsingular, and $\Sigma$ and $\Omega$ are $p \times p$ diagonal matrices: $\Sigma = \mathrm{diag}(\sigma_1, \sigma_2, \ldots, \sigma_p)$ and $\Omega = \mathrm{diag}(\mu_1, \mu_2, \ldots, \mu_p)$ with $0 \le \sigma_1 \le \sigma_2 \le \ldots \le \sigma_p < 1$ and $1 \ge \mu_1 \ge \mu_2 \ge \ldots \ge \mu_p > 0$, satisfying $\Sigma^\top \Sigma + \Omega^\top \Omega = I_p$. The *generalized singular values* $\gamma_i$ of $(A, L)$ are defined by the ratios $\gamma_i = \sigma_i / \mu_i$ for $i \in [p]$.

We prove that for the spline regression, our sketching matrix can have even smaller dimension. The following is due to [DSSW18].

**Lemma D.3.** *Let $x^* = \arg\min_{x \in \mathbb{R}^d} \|Ax - b\|_2^2 + \lambda \|Lx\|_2^2$, $A \in \mathbb{R}^{n \times d}$ and $b \in \mathbb{R}^n$. Let $U_1 \in \mathbb{R}^{n \times d}$ be the first $n$ rows of an orthogonal basis for $\begin{bmatrix} A \\ \sqrt{\lambda} L \end{bmatrix} \in \mathbb{R}^{(n+p) \times d}$. Let sketching matrix $S \in \mathbb{R}^{m \times n}$ have a distribution such that with probability $1 - \delta$,*

- $\|U_1^\top S^\top S U_1 - U_1^\top U_1\|_2 \le 1/4$,

- $\|U_1^\top (S^\top S - I)(b - Ax^*)\|_2 \le \sqrt{\varepsilon\,\mathrm{OPT}/2}$.

*Let $\widetilde{x}$ denote $\arg\min_{x \in \mathbb{R}^d} \|S(Ax - b)\|_2^2 + \lambda \|Lx\|_2^2$. Then with probability at least $1 - \delta$,*

$$\|A\widetilde{x} - b\|_2^2 + \lambda \|L\widetilde{x}\|_2^2 = (1 \pm \varepsilon)\mathrm{OPT}.$$

**Lemma D.4.** *For $U_1$ as in Lemma D.3, we have*

$$\|U_1\|_F^2 = \mathtt{sd}_\lambda(A, L).$$

---

**Algorithm 3** Dynamic Tensor Spline Regression

---

1: **data structure** DTSREGRESSION
2:
3: **members**
4:     DYNAMICTENSORTREE DTT
5:     $A_1 \in \mathbb{R}^{n_1 \times d_1}, \ldots, A_q \in \mathbb{R}^{n_q \times d_q}$
6:     $\widetilde{b} \in \mathbb{R}^m$
7: **end members**
8:
9: **procedure** INITIALIZE($A_1 \in \mathbb{R}^{n_1 \times d_1}, \ldots, A_q \in \mathbb{R}^{n_q \times d_q}, m, C_{\text{base}}, T_{\text{base}}, b \in \mathbb{R}^n$)
10:     DTT.INITIALIZE($A_1, \ldots, A_q, m, C_{\text{base}}, T_{\text{base}}$)
11:     $A_1 \leftarrow A_1, \ldots, A_q \leftarrow A_q$
12:     Let $\Pi_q \in \mathbb{R}^{m \times n}$ be the sketching matrix corresponds to DTT
13:     $\widetilde{b} \leftarrow \Pi_q b$
14: **end procedure**
15:
16: **procedure** QUERY()
17:     $M \leftarrow \text{DTT}.J_{0,0}$
18:     $\widetilde{x} \leftarrow \arg\min_{x \in \mathbb{R}^n} \|Mx - \widetilde{b}\|_2^2 + \|Lx\|_2^2$
19:     **return** $x$
20: **end procedure**

---

We are now ready to prove the main theorem.

**Theorem D.5** (Formal version of Theorem 6.4)**.** *There exists an algorithm (Algorithm 3) with the following procedures:*

- INITIALIZE($A_1 \in \mathbb{R}^{n_1 \times d_1}, \ldots, A_q \in \mathbb{R}^{n_q \times d_q}$)*: Given matrices $A_1 \in \mathbb{R}^{n_1 \times d_1}, \ldots, A_q \in \mathbb{R}^{n_q \times d_q}$, the data structure pre-processes in time $\widetilde{O}(\sum_{i=1}^q \mathrm{nnz}(A_i) + qmd + m \cdot \mathrm{nnz}(b))$.*

- UPDATE($i \in [q], B \in \mathbb{R}^{n_i \times d_i}$)*: Given an index $i \in [q]$ and an update matrix $B \in \mathbb{R}^{n \times d}$, the data structure updates in time $\widetilde{O}(\mathrm{nnz}(B) + md)$.*

- QUERY: *Let $A$ denote the Tensor product $\bigotimes_{i=1}^{q} A_i$. Query outputs a solution $\widetilde{x} \in \mathbb{R}^n$ with constant probability such the $\widetilde{x}$ is an $\varepsilon$ approximate to the Tensor Spline Regression problem i.e. $\|A\widetilde{x} - b\|_2^2 + \lambda \cdot \|L\widetilde{x}\|_2^2 \le (1+\varepsilon) \cdot \mathrm{OPT}$ where $\mathrm{OPT} = \min_{x \in \mathbb{R}^n} \|Ax - b\|_2^2 + \lambda \cdot \|Lx\|_2^2$. Query takes time $\widetilde{O}(md^{(\omega-1)} + pd^{(\omega-1)} + d^\omega)$.*

*Moreover, we have that if*

- $C_{\mathrm{base}}$ *is* COUNTSKETCH, $T_{\mathrm{base}}$ *is* TENSORSKETCH, *then* $m = \Omega(\varepsilon^{-1} q\mathtt{sd}_\lambda^2(A, L)1/\delta)$.

- $C_{\mathrm{base}}$ *is* OSNAP, $T_{\mathrm{base}}$ *is* TENSORSRHT, *then* $m = \widetilde{\Omega}(\varepsilon^{-1} q^4 \, \mathtt{sd}_\lambda(A, L) \log(1/\delta))$.

*Proof.* **Proof of Correctness for QUERY.**

To prove that it is enough to use sketching whose dimension depends on $\mathtt{sd}_\lambda(A, L)$, we just need to show that the sketching matrix related to the dynamic tree satisfies the two guarantees of Lemma D.3. We will use approximate matrix product (Def. A.3) to prove this guarantee. By Lemma D.4, we have that $\|U_1\|_F^2 = \mathtt{sd}_\lambda(A, L)$, and consider an $(\sqrt{\varepsilon}/\mathtt{sd}_\lambda(A, L), \delta)$-approximate matrix product sketching, we have that

$$
\begin{aligned}
\|U_1^\top S^\top S U_1 - U_1^\top U_1\|_2 &\le \|U_1^\top S^\top S U_1 - U_1^\top U_1\|_2 \\
&\le \sqrt{\varepsilon}\|U_1\|_F^2 / \mathtt{sd}_\lambda(A, L) \\
&= \sqrt{\varepsilon},
\end{aligned}
$$

setting $\varepsilon \le 1/2$, we obtain desired result. For the second part, we set $C = U_1$ and $D = b - Ax^*$, then

$$
\begin{aligned}
\|U_1^\top (S^\top S - I)(b - Ax^*b)\|_2 &\le \|U_1^\top (S^\top S - I)(b - Ax^*b)\|_F \\
&\le \sqrt{\varepsilon}/\mathtt{sd}_\lambda(A, L) \cdot \|U_1\|_F \|b - Ax^*\|_2 \\
&= \sqrt{\varepsilon/\mathtt{sd}_\lambda(A, L)} \sqrt{\mathrm{OPT}} \\
&\le \sqrt{\varepsilon \, \mathrm{OPT}/2}.
\end{aligned}
$$

This means that to obtain good approximation for Spline regression, it is enough to employ $(\sqrt{\varepsilon}/\mathtt{sd}_\lambda(A, L), \delta)$-approximate matrix product. The dimension follows from Lemma B.3 and B.4.

We note that even though the definition of statistical dimension in [AKK$^+$20] and ours is different, our definition is more general and the [AKK$^+$20] definition can be viewed as picking $L^\top L$ to be the identity matrix. Hence, one can easily generalize the conclusion of Lemma B.3, B.4 and B.5 to our definition of statistical dimension via the argument used in [ACW17, DSSW18, AKK$^+$20].

**Proof of INITIALIZATION.** The initialization part of Algorithm 3 is identical to the initialization part of Algorithm 2. Therefore, the time for initialization is $\widetilde{O}(\sum_{i=1}^{q} \mathrm{nnz}(A_i) + qmd + m \cdot \mathrm{nnz}(b))$ from Theorem C.1.

**Proof of UPDATE.** We just update our DTT data structure. This takes time $\widetilde{O}(\mathrm{nnz}(B) + md)$ from Theorem B.1.

**Proof of QUERY time.** As mentioned in Section 6.2, we can compute $\widetilde{x} \leftarrow \arg\min_{x \in \mathbb{R}^n} \|Mx - \widetilde{b}\|_2^2 + \|Lx\|_2^2$ by $\widetilde{x} \leftarrow (M^\top M + \lambda L^\top L)^{-1} M^\top \widetilde{b}$. We can view Query as consisting of the following 5 steps:

- Initializing $M$ takes time $\widetilde{O}(md)$ since $M \in \mathbb{R}^{m \times d}$.

- $M^\top(\widetilde{b})$, this takes $O(md)$ time because we are multiplying a $d \times m$ matrix with a $m \times 1$ vector.

- $M^\top \cdot M$, this step takes $\mathcal{T}_{\mathrm{mat}}(d, m, d)$ time as we are multiplying a $d \times m$ matrix with an $m \times d$ matrix.

- $L^\top L$, this step takes $\mathcal{T}_{\mathrm{mat}}(d, p, d)$ time as we are multiplying a $d \times p$ matrix with a $p \times d$ matrix.

- $M^\top M + \lambda L^\top L$, this takes $O(d^2)$ time as we are adding two $d \times d$ size matrices.

- $(M^\top M + \lambda L^\top L)^{-1}$, this step takes $\mathcal{T}_{\mathrm{mat}}(d, d, d)$ time as it is inverting a matrix of size $d \times d$.

- $(M^\top M + \lambda L^\top L)^{-1}(M^\top \widetilde{b})$. In this step, we are multiplying $(M^\top M + \lambda L^\top L)^{-1}$ which is a $d \times d$ matrix with a vector $M^\top \widetilde{b}$ of size $d$. This takes time $O(d^2)$.

Overall, the total running time of QUERY is

$$\widetilde{O}(md + md + \mathcal{T}_{\mathrm{mat}}(d, m, d) + \mathcal{T}_{\mathrm{mat}}(d, p, d) + d^2 + \mathcal{T}_{\mathrm{mat}}(d, d, d) + d^2)$$
$$= \widetilde{O}(md + md^{(\omega-1)} + pd^{(\omega-1)} + d^\omega + d^2)$$
$$= \widetilde{O}(md^{(\omega-1)} + pd^{(\omega-1)} + d^\omega)$$

$\square$

# E    Omitted details and proofs for Dynamic Tensor Low Rank Approximation (section 6.3)

For low rank approximation, we show it is enough to design sketching matrix whose dimension depends on $k$ instead of $d$.

**Definition E.1** (Projection Cost Preserving Sketch). Let $A \in \mathbb{R}^{n \times d}$, we say a sketching matrix $S \in \mathbb{R}^{m \times n}$ is a $k$ *Projection Cost Preserving Sketch* (PCPSketch if for any orthogonal projection $P \in \mathbb{R}^{d \times d}$ of rank $k$, we have

$$(1 - \varepsilon)\|A - AP\|_F^2 \le \|SA - SAP\|_F^2 + c \le (1 + \varepsilon)\|A - AP\|_F^2$$

where $c \ge 0$ is some fixed constant independent of $P$, but may depend on $A$ and $SA$.

For low rank approximation, we note that the optimal low rank approximation for $A$ is projecting onto the top $k$ left singular vectors. Hence, if we can prove that the sketching corresponds to the dynamic tree is a $k$-PCPSketch, then we are done. We will require a few technical tools.

**Lemma E.2** (Theorem 12 of [CEM$^+$15]). *Let $A \in \mathbb{R}^{n \times d}$. Suppose $S \in \mathbb{R}^{m \times n}$ is an $(\varepsilon, \delta, k, d, n)$-OSE and $(\varepsilon/\sqrt{k}, \delta)$-approximate matrix product, then $S$ is a $k$-PCPSketch for $A$.*

**Theorem E.3** (Formal version of Theorem 6.5). *There exists an algorithm (Algorithm 4) that has the following procedures*

- INITIALIZE($A_1 \in \mathbb{R}^{n_1 \times d_1}, \ldots, A_q \in \mathbb{R}^{n_q \times d_q}$): *Given matrices $A_1 \in \mathbb{R}^{n \times d}, \ldots, A_q \in \mathbb{R}^{n \times d}$, the data structure processes in time $\widetilde{O}(\sum_{i=1}^q \mathrm{nnz}(A_i) + qmd + m \cdot \mathrm{nnz}(b))$.*

- UPDATE($i \in [q], B \in \mathbb{R}^{n_i \times d_i}$): *Given an index $i \in [q]$ and an update matrix $B \in \mathbb{R}^{n \times d}$, the data structure updates the approximation for $A_1 \otimes \ldots \otimes (A_i + B) \otimes \ldots \otimes A_q$ in time $\widetilde{O}(\mathrm{nnz}(B) + md)$.*

- QUERY: *Let $A$ denote the tensor product $\bigotimes_{i=1}^q A_i$. The data structure outputs a rank-$k$ approximation $C$ such that*

$$\|C - A\|_F \le (1 + \varepsilon) \min_{\mathrm{rank}-k\ A'} \|A' - A\|_F.$$

*The time to output $C$ is $\widetilde{O}(md^{\omega-1})$.*

*Moreover, we have that if*

- $C_{\mathrm{base}}$ *is* COUNTSKETCH, $T_{\mathrm{base}}$ *is* TENSORSKETCH *and* $m = \Omega(\varepsilon^{-2}qk^2 1/\delta)$.

- $C_{\mathrm{base}}$ *is* OSNAP, $T_{\mathrm{base}}$ *is* TENSORSRHT *and* $m = \widetilde{\Omega}(\varepsilon^{-2}qk^2 \log(1/\delta))$.

---

**Algorithm 4** Our low rank approximation algorithm

---

1: **data structure** LowRankMaintenance
2:
3: **members**
4:     DynamicTensorTree DTT
5:     $A_1 \in \mathbb{R}^{n_1 \times d_1}, \ldots, A_q \in \mathbb{R}^{n_q \times d_q}$
6: **end members**
7:
8: **procedure** Initialize($A_1 \in \mathbb{R}^{n_1 \times d_1}, \ldots, A_q \in \mathbb{R}^{n_q \times d_q}$)
9:     DTT.Initialize($A_1, \ldots, A_q$)
10:     $A_1 \leftarrow A_1, \ldots, A_q \leftarrow A_q$
11: **end procedure**
12:
13: **procedure** Update($i \in [q], B \in \mathbb{R}^{n_i \times d_i}$)
14:     DTT.Update($i, B$)
15: **end procedure**
16:
17: **procedure** Query
18:     $M \leftarrow$ DTT.$J_{0,0}$
19:     Compute SVD of $M$ such that $M = U\Sigma V^\top$
20:     $U_k \leftarrow$ top $k$ right singular vectors of $M$
21:     **return** $A_1 \otimes \ldots \otimes A_q U_k^\top U_k$ in factored form
22: **end procedure**

---

*Proof.* **Proof of Correctness for Query.**

We first show that if we have a $k$-PCPSketch for $A$, then we can compute a matrix $C$ that is a good approximation to rank-$k$ low rank approximation. Recall that for any projection matrix, we have

$$(1 - \varepsilon)\|AP - A\|_F^2 \le \|SAP - SA\|_F^2 + c \le (1 + \varepsilon)\|AP - A\|_F^2,$$

we need to design the projection $P$. Let $SA = U\Sigma V^\top$ be the singular value decomposition of $SA$, and let $U_k \in \mathbb{R}^{k \times d}$ be the top $k$ left singular vectors of $SA$. Set $P = U_k^\top U_k$, we note that $SAU_k^\top U_k$ is the optimal rank-$k$ approximation for $SA$. On the other hand, let $Q \in \mathbb{R}^{d \times d}$ be the projection such that $AQ$ is the optimal rank-$k$ approximation for $A$. We have

$$\|SAU_k^\top U_k - SA\|_F^2 + c \le \|SAQ - SA\|_F^2 + c = (1 \pm \varepsilon)\min_{\text{rank-}k \text{ projection } P}\|AP - A\|_F^2.$$

By Lemma E.2, to obtain a $k$-PCPSketch, we need $(\varepsilon, \delta, k, d, n)$-OSE and $(\varepsilon/\sqrt{k}, \delta)$-approximate matrix product, choosing dimension according to Lemma B.3 and B.4 gives the desired result.

**Proof of Initialization.** The initialization part of Algorithm 3 is identical to the initialization part of Algorithm 2. Therefore, the time for initialization is $\widetilde{O}(\sum_{i=1}^q \text{nnz}(A_i) + qmd + m \cdot \text{nnz}(b))$ from Theorem C.1.

**Proof of Update.** We just update our DTT data structure. This takes time $\widetilde{O}(\text{nnz}(B) + md)$ from Theorem B.1.

**Proof of Query time.** We can view Query as consisting of the following 5 steps:

- Initializing $M$ takes time $\widetilde{O}(md)$ since $M \in \mathbb{R}^{m \times d}$.

- Computing the SVD of $M$ takes time $O(md^{\omega - 1})$ since $M$ is $m \times d$ matrix and $m \ge d$.

- $U_k$ can be obtained directly from the SVD computation so this step doesn't take any additional time.

- Once the algorithm has $U_k$, it can terminate as it outputs $(A_1, ..., A_q, U_k)$ in factored form.

Overall, the total running time of Query is

$$\widetilde{O}(md + md^{\omega - 1})) = \widetilde{O}(md^{\omega - 1}). \qquad \square$$