# OpenReview forum: "Dynamic Tensor Product Regression"
_NeurIPS.cc/2022/Conference — NeurIPS 2022 Accept_

### Official Review · Reviewer_uETh · 2022-07-08

**Rating:** 4
**Confidence:** 3
**Soundness:** 3 good
**Presentation:** 4 excellent
**Contribution:** 2 fair

**Summary:**

Given an input of $q$ matrices ${A_1, ..., A_q}$ (dimension of the matrices might differ) and a response vector $b$ of size equal to the product of all the input sizes from the $q$ matrices the 'tensor product regression problem' is to find the unknown vector whose dimension is equal to the product of all the dimensions from the $q$ matrices. such that it minimizes the $\ell_2$ norm of the difference vector between the response vector b and a vector which is a matrix-vector product between the Kronecker/tensor product all the input matrix and $x$ i.e., $||kr(A_1,...,A_q)x - b||_2$.
This paper studies a dynamic version of the problem, where at each time stamp some matrix $A_i$ gets updated and the goal is to quickly update the tensor product and also quickly update the solution $x$ (without recomputing it). To solve this problem the author shows how [AKK+20] tree data structure can be used in combination with known sketching techniques. They also show its application to tensor spline regression problems and tensor low rank approximation problems.

**Questions:**

I would like the authors to comment on the following questions.

- Since tensor product follows a well-defined structure a bottom-up tree like structure is quite intuitive. Apart from ensuring that all the sketch matrices are independent, can you point out some challenges or subtleties that one should be careful about while using this data structure?

- To realize the novelty, I would like to understand the most non trivial analysis in the paper. Can you explain why it is non trivial?

- How useful is the update time improvement from $O(q)$ to $O(\log(q))$ while the running time still depends on $d$ where $d = \exp(q)$?

- In the case of dynamic tensor product regression, at any time stamp, is your update only limited to $A$ or your tree can also handle updates in the response vector $b$? Also, can you handle the case when $B = -A_i$? I mean when the update simply deletes a matrix.

- There are many ways to decompose a tensor, e.g., cp, tucker, tensor train etc. A low-rank approximation based on these decompositions preserves the structural properties of tensors. Can you comment if there are any relation between these decompositions and your low rank approximation in section 8?

**Limitations:**

Yes, authors do talk about both scope of improvement and extension from the current result in section 9.

**Strengths And Weaknesses:**

Strength:
- The paper introduces a dynamic tree structure to handle tensor product updates in $O(\log(q))$ time instead of $O(q)$ time.
- The paper handles a more generalized version of regression and low rank approximation problems that depend on tensor products.
- The paper is very well written. I liked reading it.

Weakness:
- Motivation for the problem is limited. Some experimental results or highlighting the exact improvements in terms of running time/working space compared to naive techniques could help motivate up to some level.
- Once you build the tree, the majority of guarantees come readily from the existing results.
- Here d itself is exp(q), so improving the running time in terms of $q$ is less productive.

Typos/suggestions:
- From line 173, I understand that the leaf $T_i = S_i*A_i$, however in line 200 you use $T_i$ to represent your sketch matrix. If I am correct then the notational inconsistency makes it hard to follow else please fig-1 as an example to explain the outline of tree line-197-209.
- The sudden introduction of $J_{k,\ell}$ in line 203 is not clear. Again a relation to the fig-1 might be helpful.
- In the related work please discuss more about known tensor decomposition techniques and if your low rank approximation can be used to get an approximate decomposition of the known techniques.

---

> ### Author Response · Authors · 2022-08-02
> **Response to Reviewer uETh**
>
> Thank you for taking the time to read our paper and giving us detailed comments. Regarding motivation, magnitude of the improvement, and experiments, please read the general response.
>
> > To realize the novelty, I would like to understand the most non trivial analysis in the paper. Can you explain why it is non trivial?
>
> Our main contributions are formulating dynamic models for the tensor product regression and related problems and then showing that this particular set of data structures can be used for solving these problems. Once you see the solution, it might not look very novel but this is akin to the P vs NP problem. In the sense that, if you are playing a game of chess, and someone comes and tells you that moving the knight from d1 to f2 leads to check-mate, then you might think that there is nothing novel in doing this. But actually realizing that we can even do that is not obvious at all. This is a key part of where the novelty lies.
>
> >In the case of dynamic tensor product regression, at any time stamp, is your update only limited to $A$ or your tree can also handle updates in the response vector $b$?
>
> Please see the general response about this.
>
> > Also, can you handle the case when $B= -A_i$?
>
> Yes, such updates can be handled by our data structure. But it is not a very interesting case, since it would lead the tensor product matrix to be the all zeros matrix. The results for this would then just follow from the guarantees for the sketch matrix.
>
> >There are many ways to decompose a tensor, e.g., cp, tucker, tensor train etc. A low-rank approximation based on these decompositions preserves the structural properties of tensors. Can you comment if there are any relation between these decompositions and your low rank approximation in section 8?
>
> We are not very familiar with the literature on tensor decompositions. Due to paucity of time, we are unable to make any connections at this moment. If you think of any related literature, please feel free to mention them. Thank you again!

---

### Official Review · Reviewer_xUrd · 2022-07-11

**Rating:** 5
**Confidence:** 3
**Soundness:** 3 good
**Presentation:** 3 good
**Contribution:** 2 fair

**Summary:**

This paper studies a lightweight computation of a special case of linear regression where the design matrix is given by a tensor product. Specifically, it considers the case where one of the factor matrices temporally changes. The authors propose a tree-based data structure to maintain the sketch of the data so that we can keep up with the temporal changes.

**Questions:**

### Q1. Motivation
I can imagine that the tensor product regression is a fundamental problem and has several applications in machine learning. However, I have no idea about its dynamic extension. I think the authors could elaborate on this direction. More specifically, the application seems to be more limited in the problem setting the paper considered. Namely, for each time step, 1) only a single factor of the design matrix is changed, and 2) the exact location (index) of the factor that changes in time t is given. Can you introduce a few machine learning examples that your method particularly fits?

### Q2. Notations
- This is not a question but a suggestion. Although the proposed algorithm largely depends on the T-type sketch, its definition is in Appendix. It would be better to move to the main body.
- What is the role of matrix J? Is it either internal or leaf nodes? It suddenly appears in Section 5 without explanation.
- The description of Theorem 5.1 is too limited. What parameters do depend on \epsilon? What is the tradeoff between \epsilon and the compression efficiency (e.g. the sparseness of \Pi)?

**Limitations:**

See Questions.

**Strengths And Weaknesses:**

Strengths
1. (quality) Solid, reasonable data structure and algorithm are proposed.
2. (significance) The error analysis is provided.


Weaknesses
1. (significance) The motivation of the proposed method is not described. See Question 1.
2. (clarity) Some notations are not clear. See Question 2.
3. (significance) Empirical evaluations are not provided.

---

> ### Author Response · Authors · 2022-08-02
> **Response to Reviewer xUrd**
>
> We appreciate your careful reading and review. We want to address your two major questions below.
>
> Motivation of our paper: The major motivating machine learning example is the Kronecker graph, which is mentioned in the introduction section of part (line 37 - 38). We give a very detailed illustration of an example in the General Response section, we refer the reviewer to that part.
>
> Notations: We agree with your notation suggestions. We will move the $C$-type sketch and $T$-type sketch to the main body. The matrix $J$ represents the intermediate matrices that are either leaves or internal nodes. We will also add more clarifications in the final version of the paper. Regarding Theorem 5.1 and parameter $m$, we have the following 3 choices listed in the appendix: ($\epsilon$ is precision parameter and $\delta$ is failure probability):
>
> 1. $C$ is CountSketch and $T$ is TensorSketch, then m = $\Omega(\epsilon^{-2}qd^2/\delta)$.
>
> 2. $C$ is SRHT and $T$ is TensorSRHT, then m = $\tilde \Omega(\epsilon^{-2}q^4 d \log(1/\delta))$.
>
> 3. $C$ is OSNAP and $T$ is TensorSRHT, then $m=\tilde \Omega(\epsilon^{-2}qd^2 \log(1/\delta))$.
>
> We remark $d$ can be replaced by the statistical dimension of the problem. Choice 1 and 3 can be done in $\sum_{i=1}^q {\rm nnz}(A_i)$ time, while choice 2 can be achieved via the FFT algorithm. We will include these choices in the main body in the final version.

---

### Official Review · Reviewer_7iXa · 2022-07-11

**Rating:** 7
**Confidence:** 4
**Soundness:** 3 good
**Presentation:** 4 excellent
**Contribution:** 3 good

**Summary:**

The paper considers standard linear algebra problems with two twists on the input. First, the design matrix $A = \otimes_{i=1}^q A_i$ is a kronecker product of many smaller matrices, and it is these smaller matrices that are given to the algorithm. Second, it is a dynamic setting where, at every point of time, exactly one of the matrices undergoes arbitrary changes (so long as the matrix doesn't change dimension).

In this "Dynamic Tensor Product" setting, the paper studies standard L2 regression, Spline regression (i.e. generalized ridge regression), and Low-Rank Approximation in the Frobenius norm. These problems have been studied in the _static_ tensor product regime, but not the dynamic one, so the new results focus on the improved rates for _dynamic-ness_. They initialize this line of research.

They show that in this dynamic setting, by maintaining a binary tree of sketches of tensor products of $A_i$ matrices, their algorithms can achieve runtimes $q$ times better than the naive algorithm, where a new static data structure is generated at every update.

This is a theoretical paper -- there are no experiments. This paper makes no claims about the optimality of their algorithms, and only gives upper bounds.

**Questions:**

## Big Question
I have one core question about understanding the technical correctness of the paper:
> How can a single sketch matrix (e.g. on one of the leaf nodes) be used for arbitrarily many additive updates to the underlying matrix $A_i$?

Usually, my intuition is that sketching a matrix from $n \times d$ to $O(d) \times d$ works because we can preserve the exact span of the column of $A_i$ which span a $d$ dimensional subspace, so just $d$ columns suffice. (Suppose $n \gg d$) So, the sketch matrix just has to preserve these specific $d$ directions in an $n$ dimensional space. However, arbitrary additive updates can arbitrarily change the span of the columns of $A_i$, so an updated $A_i+B_i$ could span a totally unrelated column space. This paper seems to still use the exact same sketch, so that the sketch matrix has to simultaneously maintain accurate sketches for ALL possible spans of $d$ directions in an $n$ dimensional space. Usually, this would requires a sketch of size $O(n)$. "Update" (Lines 597-608) doesn't seem to generate any new randomness, so I don't see how this works.

## Middle-Size Questions
1. Can you include a discussion of the rates of $m$ in the body of the paper? Not really knowing the value of $m$ sticks out like a sore thumb.
1. Why define statistical dimension in the body of the paper if it's never used in the body of the paper? I'd pull statistical dimension into the discussion of splines in the body of the paper, to make the rate more distinct from that of L2 Regression.
1. Consider keeping $T_{mat}$ from line 609 of the paper in the rate that appears in the body of the paper, so people can tell what the runtime is if they want to use standard matrix-matrix multiplication. This can really go either way, but I've always appreciated seeing how the GEMM runtime interacts with the algo, since practical implementations of small $\omega$ algorithms don't exist yet (to the extent of my knowledge).
1. Can you include a fully elaborated runtime somewhere, without any $\tile O()$ notation. This would be fine in the appendix.
1. Theorem 8.1 (Lines 297) doesn't seem to run any faster given the value of $k$, while the static algorithm seems to benefit quiet a bit. When is the dynamic algo faster for $k=O(1)$? I'm thinking of the $O(qd^{3.5})$ runtime from above, and $O(qd \text{poly}(k/\varepsilon))$ sounds much worse.

## Small Notes
1. [Lines 106-108] Short paragraph seems unnecessary. The subsection headers too.
1. [Lines 127-130] Sarlos doesn't seem to define a ridge based definition of OSE in his paper. Be sure to cite wherever this notion came from, be it Mahoney or Musco or whoever is the author of this notion of regularized subspace embedding.
1. [Lines 123-133] None of these definitions seem to be used anywhere in the body of the paper. If they all are just used in the appendix, then just define them in the appendix.
1. [Line 141] Should be $\sigma_1$ and $\sigma_2$, not just $\sigma$
1. [144 - 146] Remove "will", move "in Appendix A" to be before the colon.
1. [157] Clarify that $b$ is fixed at the beginning of the process.
1. [171] Clarify which linear dependence on $q$ you're referring to. No runtimes have been stated yet, so this is odd to see. Also, clarify that it'll still be exponential in $q$, but that because one one of the $q$ matrices is being changed, the runtime should roughly be a $1/q$ fraction of the static algo's runtime.
1. [199] (Related to previous questions I asked), specify some rate on $m$
1. [203] What is $J_{k,\ell}$? It's a new symbol and it isn't clearly defined in this paragraph.
1. [206] Remove the comma after $S_i \in R^{m \times m^2}$
1. [207] Consider citing something for the fast matrix-vector product runtime, or specifying why this can be done quickly.
1. [231] $\Pi^q$ is not the matrix $\Pi$ multiplied to itself $q$ times (at least, I don't think it is...). So, consider using a different symbol. Why not just $\Pi$, without any $q$?
1. [279] Add "such that" before "with". Remove "such the".
1.

**Limitations:**

No concerns

**Strengths And Weaknesses:**

The paper initializes an interesting line of research, it is very clearly and well written, it has a single core observation that explains the design of their algorithm and their improved runtimes. **It's a clean and easy paper to accept, but nothing groundbreaking.** I elaborate below.

## Originality
The paper has two cores of originality:
1. Initializing the research of dynamic tensor products regression and low-rank approximation
1. Designing a dynamic data structure that can outperform repeatedly building the static data structure

The first point is stronger, since there seems to be no particular prior work on this precise problem, and because tensor product data does arise in practice (Lines 21-28). The second point is a bit weaker since the problem statement allows for updates to be *very* small -- just changing one of the $q$ base matrices, so an algorithm with updates that are $q$ times faster than rebuilding from scratch is ... believable. The way that the paper builds and updates matrices via sketches is nice and the intuition is clear, but it's hard to ascribe the construct a huge weight of novelty.

## Significance
This ties closely into the "Originality" section above. The paper's new line of research, on specifically _dynamic_ tensor product regression and low-rank approximation, is conceptually significant. I don't want to undersell how important it is to formally design a theoretical problem. It can be hard to translate an intuitive notion like "tensor product regression, but in a dynamic setting" into a formal problem statement. Formalizing this is a legitimate contribution. That said, for a paper with no experiments which introduces a new problem to solve, I feel it is very important to state why your specific definition of a dynamic setting is meaningful. Recall their definition of the dynamic setting:
> At every time point, one matrix $A_i$ in the kronecker product $A = \otimes_{i=1}^k A_i$ changes arbitrarily.

Why is this the natural setting of dynamic tensor product regression that people in downstream applications would care about? For instance, the authors specifically site how large graphs are modeled as tensor products of smaller graphs, and that these smaller graphs can change dynamically as (eg) social networks evolve (Lines 36-41). Why should I expect a small update to a social network to appear as a single arbitrary change of only one of the $A_i$ matrices, and not as a super-sparse change to all of the $A_i$ matrices? Why doesn't the response vector $b$ change with each update? While the authors argue that dynamic tensor product regression is natural, **they don't justify their precise formulation of the problem**.

For significance, it's worth examining the runtimes they achieve in clear detail. The body of the paper, i.e. before the appendix, is all written out in terms of a sketching dimension $m$, whose rates are not clearly stated (Line 199). Taking constant failure probability and constant error tolerance, we can state the runtimes of the static and dynamic algorithms for Least Square Regression pretty clearly:
- Statically solving Least Squares Regression takes $\tilde{O}(\sum_{i=1}^q \text{nnz}(A_i) + qd^{3.5})$ time
- Dynamically updating the Least Squares Solution takes $\tilde{O}(\text{nnz}(B_i) + d^{3.5})$, where $B_i$ contains the additive changes to $A_i$.
- _(the details of how I derive these rate are in paragraph are at the end of this text box)_

where $d$ is the number of columns of $A$, which is the product of the number of columns in each $A_i$, so generally a very large number even for moderate $q$.
So the dynamic runtime improvement is genuinely by a factor of $q$, but a factor of $q$ seems very small compared to $d^{3.5}$. If each base matrix $A_i$ had exactly $d_0$ columns, then $d^{3.5} = d_0^{3.5q}$. So, saving a factor of $q$ time is nice, but the runtime is still prohibitive for even moderate $q$ and small $d_0$.

**Sure, a factor of $q$ times faster is good, but it doesn't tackle the real bottleneck which should be the term that's exponential in $q$.**

Of course, for tensor-shaped problem, this may be very hard to resolve. I mostly intend this little analysis to emphasize the scale of the improvements relative to the runtime of the algorithm. It should be $q$ times faster than the static algorithm, but if both algorithms have runtime exponential in $q$, then saving a linear factor seems relatively meager.

## Clarity and Quality

The paper is well written overall. I have notes on clarity in the "Questions" box below. My only real qualm of clarity is that precise rates for $m$ should be stated in the body of the paper instead of the appendix, and the fact that $m = \Omega(qd)$ for all given constructions feels somewhat significant.

---

### Showing my work
I'm taking $m=O(qd^2)$ from Line 618 where I use CountSketch and TensorSketch with constant $\varepsilon$ and $\delta$. Then, $md^{3/2} = O(qd^{7/2})$ and $md^{\omega-1} + d^{\omega} = O(qd^{\omega+1})$. Since $7/2 = 3.5 > 3.37 = \omega+1$, the term $qd^{3.5}$ which arises from the _Initialize_ and _Update_ runtimes (Lines 244, 246) dominates the _Query_ runtime (Line 249). So, the runtime of the static method is $O(\sum \text{nnz}(A_i) + qd^{3.5})$ and the runtime of a single dynamic update is $O(\text{nnz}(B_i) + qd^{3.5})$, even including the time to run the _Query_ procedure.

---

> ### Author Response · Authors · 2022-08-02
> **Response to Reviewer 7iXa**
>
> We are extremely thankful for your careful reading and positive review. We thank you especially for your “Big question”, which we will address first.
>
> > How can a single sketch matrix (e.g. on one of the leaf nodes) be used for arbitrarily many additive updates to the underlying matrix $A_i$?
>
> The key point to note here is that the sketching matrices are decided in advance and we assume that the update we receive is oblivious to the randomness used in the sketching matrices. While thinking about this, we noticed that actually whenever we get an update for $A_i$, we can use fresh sketching matrices for all the updates from the leaf to the root. This will cost us some randomness, but in terms of time, it takes the same time as earlier (generating new sketching matrices is fast). Therefore, we can consider an updated version of our algorithm which uses fresh sketches on the path from the leaf to the root for every time step and this provides us a data structure robust to adaptive adversarial sequences of updates.
>
> Running time of Theorem 8.1: Thank you for pointing out the runtime dependence does not depend on $k$ in our algorithm. In fact, the dependence on $k$ is hidden by the parameter $m$: taking CountSketch + TensorSketch as an example, it suffices to choose $m=O(\epsilon^{-2}qk^2)$ instead of $O(\epsilon^{-2}qd^2)$. This means our algorithm initializes in time $qk^2 d^{1.5}$ and outputs a low rank approximation in time $\min( qk^2 d^2, q^2 k^4 d)$. The major bottleneck in prior best static algorithm [DJS+19] is that they naively form the Kronecker product of $q$ sketched matrices, which has to be done in very slow ${\rm poly}(k, d, 1/\epsilon)$ time, and outweighs the running time of perform SVD. Using our data structure, we provide a much faster way to process the Kronecker product part, and hence the running time of computing SVD is dominating.
>
> [DJS+19] Diao, Jayaram, Song, Sun and Woodruff. Optimal sketching for kronecker product regression and low rank approximation. NeurIPS 2019.

---

### Author Response · Authors · 2022-08-02
**General response to all reviewers**

First of all, we would like to thank all the reviewers for their detailed reviews and valuable comments, especially in these challenging times! In particular, we really appreciate the multiple comments that our paper was very clearly and well written, and also that the reviewers liked reading our paper (and also hopefully enjoyed it :)

We will further improve our paper by implementing the suggestions in the revised version of the paper. Due to time constraints, we focus mainly on responding to the reviewers here and have not yet updated the draft as we expect significant editing for the camera-ready if our paper is accepted (along with the extra allowed page). To some extent, this is also because after thinking and discussing about the “Big Question” by Reviewer 7iXa, we realized that a simple and neat modification of our data structure can significantly strengthen our guarantees and we end up solving a very important open problem which we had stated towards the end of our paper - making our data structure robust to adaptive adversarial sequences of updates. We will discuss this in more detail in the response to Reviewer 7iXa. Thank you again!!

__Motivation for updates in our model:__ Right off the bat, let us clarify that updates to the vector b are very easy to implement in our data-structure. Indeed, if we assume that the updates to b are sparse, then they are super-fast as well. Notice that the only thing we are doing with b is that instead of maintaining it directly, we are maintaining a sketch of it of size m (which you can think of as $O(qd^2)$ with $d = d_1 d_2\ldots  d_q)$. In particular, our sketch is $\Pi_q b$ (on line 13 of Algorithm 2 on page 7). This matrix-vector product computation only takes time $m \times nnz(b)$. Therefore updates of b by $\Delta b$ will also only take time $m \times nnz(\Delta b)$ to compute (just multiply and add it to the previously maintained sketch of $b$). We will clarify this in the revised version of the paper.

The other question which was raised regarding our model was to give a real-world machine learning example where such updates can be seen. As we have already mentioned in our introduction (lines 37 - 38), Kronecker graphs [LCK+10] are very popular models for real-world network data. We will give a toy example which will motivate our model of updating one matrix at a time but encourage the reviewers to see [LCK+10] for more motivation.

__Motivating Example:__ Suppose there are “n” number of male players and “n” number of female players, each of whom are represented with “d” features. These will correspond to our two matrices A and B, both in $\mathbb{R}^{n \times d}$. The matrix $A \otimes B$ will represent the matrix with $n^2$ rows each corresponding to a possible mixed doubles teams and $d^2$ features. For example, if both players have a great serve (represented on a scale of 0 to 1), then one feature would correspond to their “combined serve” score. And suppose you are using some algorithm to keep track of matches played daily (most matches are either between male-male and female-female players), then you will only update one of these matrices at a time. We are not claiming that our model can be the only valid model for Dynamic Tensor Product Regression but as we can see it is certainly one which has some potential applications. The other dynamic model (where all matrices receive small updates) is also interesting as some matches used to update the machine learning model can also be mixed-doubles matches. We don’t see how that setting can be solved more efficiently than just recomputing the entire solution from scratch (as all entries of the final tensor product matrix can be potentially affected by such changes).

__Improvement can be better?__ Multiple reviewers felt that the improvement in runtimes is not great when compared to using the static algorithm. We don’t deny this. But as we have mentioned in our paper, and also has been noted by the reviewers, a primary contribution of our paper is to formally define the Dynamic Tensor Product Regression problem and provide the first algorithm to solve it in a faster manner. The improvement is a multiplicative factor of $q$, which can be pretty large for large $q$.

__Regarding experiments:__ Our paper’s contributions are primarily theoretical. We did not feel that there would be much utility in providing experiments by using synthetic data on small instances and by comparing it to the static algorithm which was not at all designed with the dynamic setting in mind. The reviewers can certainly educate us more about this but we honestly feel that they don’t provide much scientific value in the case of our paper.

[LCK+10]  Leskovec, Chakrabarti, Kleinberg, Faloutsos, and Ghahramani. Kronecker graphs: an approach to modeling networks. JMLR, 2010.

---

### Author Response · Authors · 2022-08-08
**Reviewer-Author Discussion Period ending soon, please respond**

Hello reviewers of “Dynamic Tensor Product Regression”,

Again, we would like to thank you all for taking the time to read our paper and for providing us your very valuable feedback. The reviewer-author discussion period ends in less than 48 hours. So, if you have any further questions/comments about our responses, it would be great if you could ask them soon so that we can at least have a couple of hours to respond if needed. Thank you so much again.

Warm regards,
Authors.

---

### Meta-Review · Area_Chair_JCAT · 2022-08-30

**Recommendation:** Accept
**Confidence:** Less certain

**Metareview:**

My main concern, that was addressed by the reviewers and was not answered by the authors, is that the improvement is q times faster algorithm for an algorithm that takes time exponential in q. In addition, the missing experimental results makes this a very theoretical paper.

Still, I recommend to accept the paper due to the significance of the problem, and conditioned on the promise of the authors to update the requested changes in the final verison.



**Award:**

No

---

### Decision · Program_Chairs · 2022-09-14

Accept